# On the Effective Number of Linear Regions in Shallow Univariate ReLU Networks: Convergence Guarantees and Implicit Bias

Itay Safran[*]
Princeton University

Gal Vardi[*]
TTI-Chicago and Hebrew University

Jason D. Lee
Princeton University

## Abstract

We study the dynamics and implicit bias of gradient flow (GF) on univariate ReLU neural networks with a single hidden layer in a binary classification setting. We show that when the labels are determined by the sign of a target network with $r$ neurons, with high probability over the initialization of the network and the sampling of the dataset, GF converges in direction (suitably defined) to a network achieving perfect training accuracy and having at most $\mathcal{O}(r)$ linear regions, implying a generalization bound. Unlike many other results in the literature, under an additional assumption on the distribution of the data, our result holds even for mild over-parameterization, where the width is $\tilde{\mathcal{O}}(r)$ and independent of the sample size.

## 1 Introduction

Over-parameterized neural networks are known to be easier to train compared to their smaller counterparts, despite the resulting increase in the problem's dimensionality and the required computational resources [Daniely, 2017, Allen-Zhu et al., 2018, Safran and Shamir, 2018, Du et al., 2018, 2019, Ji et al., 2019, Zou et al., 2020, Li et al., 2020, Safran et al., 2021, Zhou et al., 2021]. However what is perhaps more surprising, is that in stark contrast to our classic understanding of generalization in machine learning models, this does not seem to degrade the generalization capabilities of the learned model in spite of the significant increase in its capacity. It is widely believed that what plays a key role in explaining this phenomenon is what is commonly referred to in the literature as *implicit bias/regularization* [Neyshabur et al., 2014, Zhang et al., 2021], where the optimization algorithm used inadvertently exhibits a bias towards empirical minimizers with a certain property that might induce better generalization. For example, such properties may include having a small norm or quasi-norm of the weights (e.g. Neyshabur [2017], Neyshabur et al. [2017], Lyu and Li [2019], Woodworth et al. [2020], Ji and Telgarsky [2020]) or a low rank solution (e.g. Razin and Cohen [2020]).

In this work, we study the dynamics and the implicit bias of GF on univariate ReLU neural networks, with the underlying assumption that the labels are determined by the sign of a target network of width $r$ (which thus changes sign between $-1$ and $1$ at most $r$ times). Our analysis reveals that under the assumption of an i.i.d., normally-distributed initialization of the weights and biases of the network, over-parameterization (i.e. the use of width strictly larger than $r$) is necessary for attaining a small population loss. Moreover, if we also assume that the width scales at least linearly (up to logarithmic terms, excluding dependence on the confidence parameter) with the length of the shortest interval on which the labels do not change sign, then with high probability over the initialization of the network and the sampling of the data, over-parameterization is sufficient for driving the empirical loss to be

---

[*]Equal contribution

36th Conference on Neural Information Processing Systems (NeurIPS 2022).

small enough so that all data instances are classified correctly. Thereafter, by analyzing the implicit bias of GF as the time $t$ tends to infinity, we show that we converge in direction (see Section 2 for a formal definition) to a network having at most $\mathcal{O}(r)$ linear regions. Since the minimal number of neurons required to express a network with an arbitrarily small loss in general is $r$, this demonstrates that the implicit bias of GF in our setting is such that optimization converges to a solution which effectively has the optimal number of neurons up to a constant factor, which also provides a clear geometric interpretation with an intuitive generalization bound. This is in contrast to norm-based results where the characterization of the implicit bias in function space is less clear. Overall, our analysis provides an end-to-end result on the learnability of univariate ReLU neural networks with respect to GF in a binary classification setting.

The remainder of this paper is structured as follows: After specifying our contributions in more detail below, we turn to discuss related work. In Section 2, we present our notations and assumptions used throughout the paper before we present our main theorem. In Section 3 we formally present our optimization result. In Section 4 we turn to analyze the implicit bias of GF in our setting. Finally, in Section 5 we show that the implicit bias leads to a generalization bound.

**Our contributions**

- We prove that when training a sufficiently wide network which is initialized appropriately (Assumption 2.1) and under a suitable assumption on the distribution of the data (Assumption 2.2), then with high probability there exists some time $t_0$ where GF attains training error at most $\frac{1}{2n}$ on a size-$n$ sample (Theorem 3.1). Our width requirement depends on the length of the shortest interval where the classificiation does not change sign, which in certain cases requires only mild over-parameterization with far fewer parameters in the model compared to observations in the dataset (see Remark 2.1).

- We show that if GF achieves training error smaller than $\frac{1}{n}$ at some time $t_0$, then it converges to zero loss and converges in direction (see Section 2 for a formal definition) to a network with at most $\mathcal{O}(r)$ linear regions (Theorem 4.2). Since this result provides a simple and intuitive geometric interpretation to the implicit bias of GF in our setting, it readily translates to a generalization bound using a standard argument (Corollary 5.1). We point out that unlike our optimization guarantee, this result holds regardless of the initialization of the network.

- Finally, we combine our optimization and generalization results to derive our main theorem (Theorem 2.3), which establishes an end-to-end learnability result for GF in our setting. This indicates that a wide model facilitates optimization, yet at the same time implicit bias prevents us from overfitting, even if the architecture we train is far wider than what is necessary. The result holds in the *rich regime*, and thus provides a guarantee which goes beyond the analysis achieved using NTK-based results (see Remark 2.3).

- As an additional contribution, we show (under our assumption on the initialization) that width at least $1.3r$ is necessary for GF to attain population loss less than an absolute constant over a particular target function (Theorem 3.2). Along with our previous results, this shows that over-parameterization is not just sufficient, but also necessary for successful learning.

We now turn to discuss some of the related work in the literature which is most relevant to ours in more detail.

**Related work**

**Implicit bias in neural networks.** The literature on the implicit bias in neural networks has rapidly expanded in recent years, and cannot be reasonably surveyed here (see Vardi [2022] for a survey). In what follows, we discuss only results which apply to depth-2 ReLU networks.

By Lyu and Li [2019], Ji and Telgarsky [2020] homogeneous neural networks (and specifically depth-2 ReLU networks) trained with exponentially-tailed classification losses converge in direction to a KKT point of the maximum-margin problem. Our analysis of the implicit bias relies on this result. We note that the aforementioned KKT point may not be a global optimum (see a discussion in Section 4). For depth-2 ReLU networks trained with the square loss there are no known guarantees on the implicit bias (cf. Vardi and Shamir [2021], Timor et al. [2022]).

Several works in recent years studied the implication of minimizing the $\ell_2$ norm of the weights on the function space in depth-2 univariate ReLU networks. In Savarese et al. [2019] and Ergen and Pilanci [2021a] it is shown that a minimal-norm fit for a sample is given by the linear spline interpolation (i.e., a "connect-the-dots" function). In such linear spline interpolation the number of linear regions is small. The former work considered only regression, while the latter considered both regression and classification. Note that margin-maximization is equivalent to norm-minimization with margin at least 1. Thus, our result can also be viewed as an analysis of the implication of the bias towards norm-minimization on the learned function. We emphasize three important differences between the results from Savarese et al. [2019], Ergen and Pilanci [2021a] and ours:

1. As we already mentioned, the result of Lyu and Li [2019], Ji and Telgarsky [2020] implies a certain bias towards margin maximization, but it does not guarantee convergence to a global optimum (or even to a local optimum) of the maximum-margin problem. The only guarantee is that GF converges to a KKT point. Our result relies only on convergence to such a KKT point, and (unlike Savarese et al. [2019] and Ergen and Pilanci [2021a]) it does not assume convergence to a global optimum. As a result, we are able to obtain provable generalization bounds for GF.

2. In Savarese et al. [2019] and Ergen and Pilanci [2021a] it is shown that the linear spline interpolation minimizes the weights' norms. However, they also show that it is not a unique minimizer. Thus, in addition to the linear spline interpolation there are also other networks that fit the training set and minimize the norms. Therefore, even under the assumption that the weights' norms are minimized, their results do not guarantee convergence to a function with a small number of linear regions (as in our result).

3. Savarese et al. [2019] and Ergen and Pilanci [2021a] consider norm-minimization of the weights without the bias terms, while the implicit bias towards margin-maximization due to Lyu and Li [2019], Ji and Telgarsky [2020] (which we rely on) is w.r.t. all the parameters, including the bias terms. Hence, the implicit bias in depth-2 ReLU networks with exponentially-tailed losses does not minimize the norms in the sense considered in Savarese et al. [2019], Ergen and Pilanci [2021a].

We note that Ergen and Pilanci [2021b] showed that linear spline interpolators minimize the norms also in deep univariate networks. The result from Savarese et al. [2019] was extended to multi-variate functions in Ongie et al. [2019]. Parhi and Nowak [2020] studied the relation between norm minimization and spline interpolation for a broader family of activation functions. Hanin [2021] gave a geometric characterization of all depth-2 univariate ReLU networks with a single linear unit, that minimize the $\ell_2$ norm of the weights (excluding bias terms) and interpolate a given dataset (in a regression setting). Blanc et al. [2020] studied the relation between the implicit bias of SGD in depth-2 univariate ReLU networks (in a regression setting) and the number of convexity changes of the learned network. Maennel et al. [2018] showed that for a given training dataset there are only finitely many functions that GF with small initialization may converge to in depth-2 ReLU networks, independent of the network size. Chizat and Bach [2020] studied the dynamics of GF on infinite width depth-2 networks with exponentially-tailed losses and showed bias towards margin maximization w.r.t. a certain function norm known as the variation norm. Phuong and Lampert [2020] studied the implicit bias in depth-2 ReLU networks trained on orthogonally separable data.

**Convergence of gradient methods under extreme over-parameterization.** In recent years, many theoretical works have focused on providing convergence guarantees for training depth-2 neural networks with non-linear activations. Andoni et al. [2014] provide a convergence guarantee for learning polynomials of degree $r$ in $d$-dimensional space in a regression setting using networks of width roughly $d^{2r}$. Since their architecture excludes bias terms which can be simulated by incrementing the input dimension by 1, their result in fact requires width $2^{2r}$ in our univariate setting, whereas our width requirement is typically much milder. Following the success of the NTK [Jacot et al., 2018], a spate of papers provided convergence guarantees when training using GD (e.g. [Allen-Zhu et al., 2018, Du et al., 2018, 2019, Ji et al., 2019, Zou et al., 2020]). The main difference that sets our work apart is that our width requirement is given in terms of the complexity of the teacher network, irrespective of the sample size $n$, whereas these works require that the width scales polynomially with $n$, which could be significantly larger. Moreover, as mentioned earlier, such results operate in the lazy regime where the features that are learned are dictated mainly by the initialization rather than the training process, whereas our analysis enters the rich regime once the

loss becomes sufficiently small, and provides a result that goes beyond NTK-based analyses. On the flip side, our analysis only holds for binary classification in the one-dimensional setting. Similarly to us, Soltanolkotabi et al. [2018] provide convergence guarantees by establishing that the objective function satisfies the PL-condition (see Polyak [1963]), however unlike our optimization guarantee and similarly to previously discussed works, their result requires that the network has more trainable parameters than data instances. Chizat et al. [2019] establish that by scaling a model appropriately, we can effectively interpolate between the lazy and the rich regime. While we use this observation in our optimization result to drive the loss to become sufficiently small in the first stage of optimization, our implicit bias result nevertheless operates in the rich regime regardless of this scaling.

**Teacher-student setting and mild over-parameterization.**    In this paper, we assume that the labels of the data are determined by the sign of a teacher network of width $r$. Such a similar teacher-student setting but for a regression problem allowed the study of mild over-parameterization in quite a few recent works. Safran and Shamir [2018], Arjevani and Field [2020, 2021] show the existence of spurious (non-global) local minima in the loss landscape in this setting. Other works provide certain recovery guarantees; assuming absolute value activations, Li et al. [2020] provide a global convergence guarantee to loss at most $o(1/r)$, and Safran et al. [2021], Zhou et al. [2021] provide local convergence guarantees for ReLU or absolute value activations. These works require width at least $\text{poly}(r)$ for convergence, whereas in our setting we show that width $\tilde{\mathcal{O}}(r)$ suffices in certain cases, and that over-parameterization is also necessary for successful optimization under our assumptions. While our results might superficially seem to provide stronger guarantees, we stress that the seemingly stronger bounds we derive are made possible in part due to the different assumptions made which include a univariate domain with biases compared to a multivariate domain with no biases nor output layer weights, thus highlighting the difference between the two architectures. In light of this, we argue that the bounds in these results are not directly comparable to ours.

## 2    Preliminaries and main result

**Notations.**    We use bold-face letters to denote vectors, e.g., $\mathbf{x} = (x_1, \ldots, x_d)$. For $\mathbf{x} \in \mathbb{R}^d$ we denote by $\|\mathbf{x}\|$ the Euclidean norm. We denote by $\mathbb{1}[\cdot]$ the indicator function, for example $\mathbb{1}[t \geq 5]$ equals 1 if $t \geq 5$ and 0 otherwise. We denote $\text{sign}(z) = 1$ if $z > 0$ and $-1$ otherwise. For an integer $d \geq 1$ we denote $[d] = \{1, \ldots, d\}$. We use standard asymptotic notation $\mathcal{O}(\cdot)$ to hide constant factors. A function $f : D \to \mathbb{R}$ which is twice continuously differentiable in a domain $D \subseteq \mathbb{R}^d$ is said to satisfy the *PL-condition* if there exists $\lambda > 0$ such that $\frac{1}{2}\|\nabla f(\mathbf{x})\|^2 \geq \lambda(f(\mathbf{x}) - f^*)$, where $f^* := \inf_{\mathbf{x}} f(\mathbf{x})$.

**Neural networks.**    The ReLU activation function is defined by $\sigma(z) = \max\{0, z\}$. In this work we consider depth-2 ReLU neural networks with input dimension 1. Formally, a depth-2 network $\mathcal{N}_{\boldsymbol{\theta}}$ of width $k$ is parameterized by $\boldsymbol{\theta} = [\mathbf{w}, \mathbf{b}, \mathbf{v}]$ where $\mathbf{w}, \mathbf{b}, \mathbf{v} \in \mathbb{R}^k$, and for every input $x \in \mathbb{R}$ we have

$$\mathcal{N}_{\boldsymbol{\theta}}(x) = \sum_{j \in [k]} v_j \sigma(w_j \cdot x + b_j). \tag{1}$$

We sometimes view $\boldsymbol{\theta}$ as the vector obtained by concatenating the vectors $\mathbf{w}, \mathbf{b}, \mathbf{v}$. Thus, $\|\boldsymbol{\theta}\|$ denotes the $\ell_2$ norm of the vector $\boldsymbol{\theta}$. We denote

$$\Phi(\boldsymbol{\theta}; x) := \mathcal{N}_{\boldsymbol{\theta}}(x).$$

Given a network $\mathcal{N}_{\boldsymbol{\theta}}(x)$ as above, we refer to the set of its non-differentiable points (w.r.t. the variable $x$) as its breakpoints.

**Gradient flow (GF) and implicit bias.**    Let $S = \{(x_i, y_i)\}_{i=1}^n \subseteq \mathbb{R} \times \{-1, 1\}$ be a binary classification training dataset. Let $\Phi(\boldsymbol{\theta}; \cdot) : \mathbb{R} \to \mathbb{R}$ be a neural network parameterized by $\boldsymbol{\theta}$. For a loss function $\ell : \mathbb{R} \to \mathbb{R}$ the *empirical loss* of $\Phi(\boldsymbol{\theta}; \cdot)$ on the dataset $S$ is

$$\mathcal{L}(\boldsymbol{\theta}) := \frac{1}{n} \sum_{i=1}^n \ell(y_i \Phi(\boldsymbol{\theta}; x_i)). \tag{2}$$

We focus on the exponential loss $\ell(q) = e^{-q}$ and the logistic loss $\ell(q) = \log(1 + e^{-q})$.

We consider GF on the objective given in Eq. (2). This setting captures the behavior of GD with an infinitesimally small step size. Let $\boldsymbol{\theta}(t)$ be the trajectory of GF. Starting from an initial point $\boldsymbol{\theta}(0)$, the dynamics of $\boldsymbol{\theta}(t)$ are given by the differential equation $\frac{d\boldsymbol{\theta}(t)}{dt} \in -\partial^\circ \mathcal{L}(\boldsymbol{\theta}(t))$. Here, $\partial^\circ$ denotes the *Clarke subdifferential*, which is a generalization of the derivative for non-differentiable functions (see Appendix A for a formal definition). We say that a trajectory $\boldsymbol{\theta}(t)$ *converges in direction* to $\boldsymbol{\theta}^*$ if $\lim_{t\to\infty} \frac{\boldsymbol{\theta}(t)}{\|\boldsymbol{\theta}(t)\|} = \frac{\boldsymbol{\theta}^*}{\|\boldsymbol{\theta}^*\|}$.

## 2.1 Assumptions

Our main result holds under the following assumption on the initialization of the weights of the network.

**Assumption 2.1** (Network initialization).

- *The weights and biases $w_i, b_i$ of each neuron $i \in [k]$ in the hidden layer are i.i.d. and satisfy*

$$w_i, b_i \sim \mathcal{N}(0, \sigma_h^2).$$

- *The weights $v_i$, $i \in [k]$ of the output neuron are i.i.d. and satisfy*

$$v_i \sim \mathcal{N}(0, \sigma_o^2).$$

We point out that our particular choice of normally distributed weights is not essential, and that our results will also hold for example under the assumption of uniformly distributed weights, but with a slightly different proof and constants in the resulting bounds. To facilitate our analysis, we make the following assumptions on the distribution of the data and its corresponding labels.

**Assumption 2.2** (Data distribution). *There exist a natural $r \geq 1$ and real $R \geq 1$ and $C, \rho > 0$ such that following hold:*

- *There exists a depth-2 ReLU network $\mathcal{N}^*$ of width $r$ such that the examples $(x, y)$ of the data satisfy $x \sim \mathcal{D}$ and $y = \text{sign}(\mathcal{N}^*(x))$.*

- *The density of $\mathcal{D}$ denoted by $\mu$ satisfies $\mu(x) = 0$ for all $x \notin [-R, R]$.*

- $\sup_x \mu(x) \leq C$.

- *$\rho > 0$ is the length of the shortest interval $I \subseteq [-R, R]$ such that $\text{sign}(\mathcal{N}^*(x))$ is the same for all $x \in I$, and for all $I \subset I'$, there exist $x, x' \in I'$ such that $\text{sign}(\mathcal{N}^*(x)) \neq \text{sign}(\mathcal{N}^*(x'))$.*

We remark that the above assumption is mostly mild. The main non-trivial requirement is that the distribution is compactly-supported, however the last two assumptions always hold for some $C, \rho > 0$ if $\mu$ is continuous on $\mathbb{R}$ for example, and any target function that changes sign at most $r$ times can be expressed by a network of width $r$.[2]

## 2.2 Main result

Having stated our assumptions, we now turn to present our main theorem in this paper.

**Theorem 2.3.** *Under Assumptions 2.1 and 2.2, given any $\varepsilon, \delta \in (0, 1)$, suppose that the following hold*

$$n \geq C_0 \cdot \frac{r \log(1/\varepsilon) + \log(2/\delta)}{\varepsilon}, \qquad k \geq 6144 \cdot \frac{R^4 \log\left(\frac{48r}{\delta}\right)}{\rho},$$

$$\sigma_h \geq 8400 \cdot \frac{n^2 C R^{3.5} \sqrt{rk}}{\delta\rho}, \qquad \sigma_o \leq \frac{1}{4kR\sigma_h \log\left(\frac{12k}{\delta}\right)},$$

*where $C_0 > 0$ is a universal constant. Then with probability at least $1 - \delta$ over the randomness in the initialization of the network and the sampling of a size-$n$ dataset, GF converges to zero loss,*

---

[2]We also remark that the assumption $R \geq 1$ is only for simplicity of presentation, since our results also hold for any $R < 1$ with the same bounds we get when plugging $R = 1$.

and converges in direction to $\boldsymbol{\theta}^*$ such that the network $\mathcal{N}_{\boldsymbol{\theta}^*}$ has at most $32r + 67$ linear regions and satisfies

$$\mathbb{P}_{x \sim \mathcal{D}}\left[\text{sign}(\mathcal{N}_{\boldsymbol{\theta}^*}(x)) \neq \text{sign}(\mathcal{N}^*(x))\right] \leq \varepsilon.$$

Our theorem is a result of breaking the proof into two different stages and combining them using a simple union bound. Specifically, at the first stage we use Theorem 3.1, which establishes that the empirical loss drops below $\frac{1}{n}$ with high probability; and in the second stage we use Corollary 5.1 to argue that the implicit bias takes effect once the empirical loss is sufficiently small which results in a generalization bound. The theorem suggests two interesting implications: (i) We may train an arbitrarily wide network without risk of overfitting since the implicit bias of GF dictates that we converge to a model with low capacity;[3] and (ii), we further gain a sample complexity bound which is independent of $\rho$ and $R$. We also note that only the direction of $\boldsymbol{\theta}$ affects the classification and the number of linear regions in $\mathcal{N}_{\boldsymbol{\theta}}$, and that the scale of $\boldsymbol{\theta}$ is not important here. Namely, for every $\boldsymbol{\theta}$, $x$ and every $\alpha > 0$ we have $\mathcal{N}_{\alpha\boldsymbol{\theta}}(x) = \alpha^2 \mathcal{N}_{\boldsymbol{\theta}}(x)$, and thus the scale of $\boldsymbol{\theta}$ affects only the scale of the outputs of $\mathcal{N}_{\boldsymbol{\theta}}$ and not the classification nor the partition to linear regions and therefore nor does it affect the generalization properties of $\boldsymbol{\theta}$.

Lastly, we remark that our initialization scheme used in Theorem 2.3 is somewhat unorthodox, in the sense that we require the hidden layer to have a rather large variance which scales polynomially with the sample size. While in practice it is more common that the weights in the hidden layer have a smaller variance (e.g. Glorot and Bengio [2010], He et al. [2015]), our scaling prevents the breakpoints of the neurons in the trained network to move too much before we are able to decrease the training error sufficiently, and the impact of the magnitude by which we scale the hidden layer upon initialization on the dynamics of GF was studied in a similar univariate regression setting [Williams et al., 2019, Sahs et al., 2020]. We now conclude the discussion of our main result with the following remarks on the setting studied in our paper.

**Remark 2.1** (Mild vs. extreme over-parameterization). *In this paper, we make a distinction between what we call the* mild over-parameterization *regime, where the required width of the network $k$ scales with $r$ but not with the sample size $n$; and the* extreme over-parameterization *regime, where the width $k'$ of the network exceeds $n$. Under this distinction, for sufficiently large $n$, we will always have that $k \ll n < k'$, and thus $k \ll k'$.*

**Remark 2.2** (GF vs. GD). *It is important to stress that our results hold for GF which ignores computational considerations and does not necessarily imply the convergence of GD. For this reason, it is of utmost importance to generalize our results to hold for GD rather than just GF. That being said, there is some recent evidence suggesting that at least in certain cases, positive results on GF may be translated to GD [Elkabetz and Cohen, 2021]. Moreover, at the very least, Theorem 3.1 can indeed be generalized to hold for GD (see discussion after the theorem statement). In any case, for the sake of coherence we focus in this paper on GF, and we leave generalizations for GD as an important future work direction.*

**Remark 2.3** (Rich vs. lazy regime). *Our optimization analysis operates in the lazy regime where the hidden layer does not move much and most of the learning is performed in the output neuron. However, once our analysis goes into the second stage where the implicit bias takes effect, we essentially move into the rich (aka the feature-learning) regime, where redundant features (i.e. excess neurons) are being effectively discarded at the limit $t \to \infty$. In light of this, as was discussed in the related work section, our result provides a guarantee which goes beyond the analysis achieved using NTK-based results.*

## 3 Over-parameterization leads to small empirical loss

In this section, we analyze the dynamics of GF on the objective defined in Eq. (2). Our main contribution is to establish that sufficient over-parameterization guarantees that GF leads to a point with empirical loss which is sufficiently small. Formally, we present the following theorem.

---

[3]Note that in our univariate setting we can deduce $r$ by analyzing the data and possibly estimate the minimal required width for attaining small training error using our derived bounds, however in more general settings (e.g. the multivariate case) $r$ may not be easily deduced from the dataset, which might prompt us to train the widest network possible given available computational resources.

**Theorem 3.1.** *Under Assumptions 2.1 and 2.2, given any $\delta \in (0,1)$, suppose that the following hold*

$$k \geq 6144 \cdot \frac{R^4 \log\left(\frac{24r}{\delta}\right)}{\rho}, \quad \sigma_h \geq 4200 \cdot \frac{n^2 C R^{3.5} \sqrt{rk}}{\delta \rho} \quad and \quad \sigma_o \leq \frac{1}{4kR\sigma_h \log\left(\frac{6k}{\delta}\right)}. \quad (3)$$

*Then with probability at least $1 - \delta$ over the randomness in the initialization of the network and the sampling of a size-$n$ dataset, there exists time $t_0$ such that GF initialized from $\boldsymbol{\theta}(0)$ reaches a point $\boldsymbol{\theta}(t_0)$ satisfying*

$$\mathcal{L}(\boldsymbol{\theta}(t_0)) \leq \frac{1}{2n}.$$

The above theorem essentially requires that we use width which is proportional to $1/\rho$ up to logarithmic factors. If $\rho = \Omega(1/r)$, then this requires that we over-parameterize by a multiplicative constant up to logarithmic factors. We remark that our result can also be adapted to hold for GD rather than GF with a polynomial number of iterations.[4] In any case, as discussed in Remark 2.2, we stress that our focus here is to show that GF attains sufficiently small loss so that our implicit bias analysis takes effect, and we leave generalizations for GD and milder width requirements for future work.

The proof of the above theorem, which appears in Appendix C, relies on over-parameterizing sufficiently to the extent of having at least three breakpoints on each constant segment where the data does not change classification, and four additional neurons that are active on all the data instances. The key observation is that under such over-parameterization, we can identify a direction in weight space which moves the current network configuration in a manner which strictly decreases the objective value. This allows us to establish that the objective function satisfies the PL-condition locally in a neighborhood around our initialization. Finally, by bounding the length of the trajectory of GF, we show that the objective value decreases to $\frac{1}{2n}$ before we can escape the neighborhood in which the PL-condition is satisfied.

Interestingly, Theorem 3.1 already implies a generalization bound if the sample size is sufficiently larger than the degrees of freedom in the student network. Nevertheless, such an approach alone is not capable of obtaining a sample complexity which is independent of $\rho$ and $R$ (since the width of the student network and thus also its capacity scale with these parameters, implying a generalization bound that explicitly depends on them). Moreover, understanding the implicit bias of GF is of independent interest, even if we ignore the improvement it provides to the sample complexity.

It is natural to explore what is the minimal amount of over-parameterization required for attaining a small loss in our setting. A modest requirement is that we are able to make the generalization error arbitrarily small given a sufficiently large sample. It is thus interesting to present the following theorem, which establishes that under Assumption 2.1, GF is not capable of attaining loss below an absolute constant unless $k \geq \lfloor 1.3r \rfloor$, regardless of the sample size.

**Theorem 3.2.** *Define the population loss of a network with weights $\boldsymbol{\theta}$ w.r.t. a distribution $\mathcal{D}$ and teacher network $\mathcal{N}^*(\cdot)$ as*

$$\mathcal{L}_{\mathcal{D}}(\boldsymbol{\theta}) := \mathbb{E}_{x \sim \mathcal{D}} \left[ \ell(\mathcal{N}_{\boldsymbol{\theta}}(x) \cdot \text{sign}(\mathcal{N}^*(x))) \right].$$

*Let $\alpha \geq 1$ and suppose that an architecture as in Eq. (1), having width $k = \lfloor \alpha r \rfloor$ is initialized according to Assumption 2.1. Then there exists a distribution $\mathcal{D}$ and $\mathcal{N}^*(\cdot)$ of width $r$ such that for any time $t \geq 0$ and any $n \geq 1$, for a size-$n$ sample drawn from $\mathcal{D}$ and labeled by $\text{sign}(\mathcal{N}^*(\cdot))$, with probability at least $0.25$ over the initialization of the network, GF trained on Eq. (2) attains population loss at least*

$$\mathcal{L}_{\mathcal{D}}(\boldsymbol{\theta}(t)) \geq \frac{1}{4} \left( 1 - 0.75\alpha \right).$$

Thus, under the theorem's assumptions, over-parameterization by a constant factor is required. We refer the reader to Appendix D for the construction of $\mathcal{D}$ and $\mathcal{N}^*(\cdot)$, some further discussion, and the full proof of this lower bound.

---

[4]To show this, one would need to bound the length of the trajectory of GD for objectives that satisfy the PL-condition locally. See Appendix C for further detail.

# 4 The implicit bias of GF

In this section, we show that GF converges to networks where the number of linear regions is minimal up to a constant factor. We first give some required background and discuss an important result on the implicit bias which applies to depth-2 ReLU networks, and then state our result.

## 4.1 Required background

The following theorem gives an important characterization of the implicit bias in depth-2 ReLU networks:

**Theorem 4.1** (Lyu and Li [2019], Ji and Telgarsky [2020]). *Let $\Phi(\boldsymbol{\theta}; \cdot)$ be a depth-2 ReLU neural network parameterized by $\boldsymbol{\theta}$. Consider minimizing either the exponential or the logistic loss over a binary classification dataset $\{(x_i, y_i)\}_{i=1}^{n}$ using GF. Assume that there exists time $t_0$ such that*

$$\mathcal{L}(\boldsymbol{\theta}(t_0)) < \frac{1}{n},$$

*namely, $y_i \Phi(\boldsymbol{\theta}(t_0); x_i) > 0$ for every $x_i$. Then, GF converges in direction to a first order stationary point (KKT point) of the following maximum margin problem in parameter space:*

$$\min_{\boldsymbol{\theta}} \frac{1}{2} \|\boldsymbol{\theta}\|^2 \quad s.t. \quad \forall i \in [n] \;\; y_i \Phi(\boldsymbol{\theta}; x_i) \geq 1 . \tag{4}$$

*Moreover, $\mathcal{L}(\boldsymbol{\theta}(t)) \to 0$ and $\|\boldsymbol{\theta}(t)\| \to \infty$ as $t \to \infty$.*

We note that the above theorem holds for the more general case of homogeneous neural networks (in parameter space), but for this work it suffices to consider depth-2 networks, which are indeed homogeneous. Note that in ReLU networks Problem (4) is non-smooth. Hence, the KKT conditions are defined using the Clarke subdifferential. See Appendix A for more details of the KKT conditions. Theorem 4.1 characterized the implicit bias of GF with the exponential and the logistic losses for depth-2 ReLU networks. Namely, even though there are many possible directions $\frac{\boldsymbol{\theta}}{\|\boldsymbol{\theta}\|}$ that classify the dataset correctly, GF converges only to directions that are KKT points of Problem (4). We note that such a KKT point is not necessarily a global/local optimum (cf. Vardi et al. [2021]). Thus, under the theorem's assumptions, GF *may not* converge to an optimum of Problem (4), but it is guaranteed to converge to a KKT point. This is demonstrated in the following example for the case of depth-2 univariate networks, which is our focus.

**Example 1.** *Let $\mathcal{N}_{\boldsymbol{\theta}}$ be a depth-2 univariate network of width 2, namely,*

$$\mathcal{N}_{\boldsymbol{\theta}}(x) = v_1 \sigma(w_1 x + b_1) + v_2 \sigma(w_2 x + b_2).$$

*Let $S = \{(x_1, y_1), (x_2, y_2)\}$ be a size-2 dataset such that $x_1 = 4$, $x_2 = -4$ and $y_1 = y_2 = 1$. Suppose that we train $\mathcal{N}_{\boldsymbol{\theta}}$ on the dataset $S$ using GF with the exponential or the logistic loss, and that the initialization $\boldsymbol{\theta}(0)$ is such that $b_1 = v_1 = 1$ and $w_1 = w_2 = b_2 = v_2 = 0$. Note that*

$$\mathcal{L}(\boldsymbol{\theta}(0)) = \frac{1}{2} \cdot 2\ell(1) < \frac{1}{2}$$

*(for both the exponential and the logistic loss). Hence, by Theorem 4.1 GF converges to zero loss, and converges in direction to a KKT point $\boldsymbol{\theta}^*$ of Problem (4). By observing the gradient $\nabla_{\boldsymbol{\theta}} \mathcal{L}(\boldsymbol{\theta})$ it is not hard to show that for every time $t$ we have $w_2(t) = b_2(t) = v_2(t) = 0$, namely, the second neuron remains inactive, and we have $\boldsymbol{\theta}^* = \boldsymbol{\theta}(0)$ (see Appendix B for details). However, $\boldsymbol{\theta}^*$ is not a global optimum of Problem (4). Indeed, consider $\boldsymbol{\theta}'$ such that $w_1' = v_1' = v_2' = \frac{1}{2}$, $w_2' = -\frac{1}{2}$, and $b_1' = b_2' = 0$. Then, $y_i \mathcal{N}_{\boldsymbol{\theta}'}(x_i) = 1$ for all $i \in \{1, 2\}$, and we have $\|\boldsymbol{\theta}'\|^2 = 1 < 2 = \|\boldsymbol{\theta}^*\|^2$.*

The above example implies that although GF has a certain bias towards margin maximization (as shown in Theorem 4.1), it may not maximize the margin. Hence, we cannot obtain margin-based generalization bounds based on the bias towards margin maximization.

## 4.2 Characterization of the implicit bias

We now state our main result on the implicit bias:

**Theorem 4.2.** *Let $S = \{(x_i, y_i)\}_{i=1}^n \subseteq \mathbb{R} \times \{-1, 1\}$ be a dataset such that $x_1 < \ldots < x_n$, and for all $i \in [n]$ we have $y_i \mathcal{N}^*(x_i) > 0$, where $\mathcal{N}^* : \mathbb{R} \to \mathbb{R}$ is a depth-2 ReLU network of width $r$. Note that $|\{i \in [n-1] : y_i \neq y_{i+1}\}| \leq r$. Consider GF on a depth-2 neural network $\mathcal{N}_{\boldsymbol{\theta}}$ w.r.t. the dataset $S$. Assume that there exists time $t_0$ such that*

$$\mathcal{L}(\boldsymbol{\theta}(t_0)) < \frac{1}{n}.$$

*Then, GF converges to zero loss, and converges in direction to a KKT point $\boldsymbol{\theta}^*$ of Problem (4), such that the network $\mathcal{N}_{\boldsymbol{\theta}^*}$ has at most $32r + 67$ linear regions.*

Since the labels in the dataset $S$ may switch sign $r$ times, then a network that correctly classifies $S$ must contain at least $r$ linear regions. Hence, the theorem implies that GF minimizes the number of linear regions up to a constant factor. We remark that the constants 32 and 67 in the above result can be improved, but we preferred here a simpler proof over a tighter bound.

The formal proof of the theorem is given in Appendix E. Below we discuss the high-level approach. By Theorem 4.1, if there exists time $t_0$ such that $\mathcal{L}(\boldsymbol{\theta}(t_0)) < \frac{1}{n}$ then GF converges to zero loss, and converges in direction to a KKT point of Problem (4). We denote

$$\mathcal{N}_{\boldsymbol{\theta}}(x) = \sum_{j \in [k]} v_j \sigma(w_j x + b_j).$$

Assume that $\mathcal{N}_{\boldsymbol{\theta}}$ satisfies the KKT conditions of Problem (4). Thus, there are $\lambda_1, \ldots, \lambda_n \geq 0$ such that for every $j \in [k]$ we have

$$w_j = \sum_{i \in [n]} \lambda_i \frac{\partial}{\partial w_j} (y_i \mathcal{N}_{\boldsymbol{\theta}}(x_i)) = \sum_{i \in [n]} \lambda_i y_i v_j \sigma'_{i,j} x_i \, , \tag{5}$$

where $\sigma'_{i,j}$ is a subgradient of $\sigma$ at $w_j \cdot x_i + b_j$, and $\lambda_i = 0$ if $y_i \mathcal{N}_{\boldsymbol{\theta}}(x_i) \neq 1$. Likewise, we have

$$b_j = \sum_{i \in [n]} \lambda_i \frac{\partial}{\partial b_j} (y_i \mathcal{N}_{\boldsymbol{\theta}}(x_i)) = \sum_{i \in [n]} \lambda_i y_i v_j \sigma'_{i,j} \, . \tag{6}$$

In the proof we show using a careful analysis of Eqs. (5) and (6) that in an interval $[x_p, x_q]$ where the labels do not switch sign (i.e., $y_p = y_{p+1} = \ldots = y_q$) the network $\mathcal{N}_{\boldsymbol{\theta}}$ has a constant number of kinks. Since the labels switch sign at most $r$ times then we are able to conclude that $\mathcal{N}_{\boldsymbol{\theta}}$ has $\mathcal{O}(r)$ kinks as required.

## 5 Implicit bias leads to a generalization bound

Consider a depth-2 teacher ReLU network $\mathcal{N}^* : \mathbb{R} \to \mathbb{R}$ of width $r$. By Theorem 4.2, if GF reaches a sufficiently small loss at some time $t_0$, it converges in direction to a network $\mathcal{N}'$ with $\mathcal{O}(r)$ linear regions that classifies the training dataset correctly. Thus, even if the width $k$ of the learned network is extremely large, the result in Theorem 4.2 guarantees that GF converges to a network with a small number of linear regions. Consider the function $f' : \mathbb{R} \to \{-1, 1\}$ defined by $f'(x) = \text{sign}(\mathcal{N}'(x))$. Since $\mathcal{N}'$ has $\mathcal{O}(r)$ linear regions then $f'$ can be expressed by a polynomial threshold function of degree $\mathcal{O}(r)$. That is, we have $f'(x) = \text{sign}(p'(x))$ where $p'$ is a polynomial of degree $\mathcal{O}(r)$. Since $\mathcal{N}'$ attains 100% classification accuracy on the size-$n$ training dataset, then we can view GF as empirical risk minimization (ERM) w.r.t. the 0-1 loss over a class of degree-$\mathcal{O}(r)$ polynomial threshold functions in the realizable setting. The VC-dimension of this class is $\mathcal{O}(r)$, and thus we have the following generalization bound w.r.t. the 0-1 loss (see, e.g., Theorem 6.8 in Shalev-Shwartz and Ben-David [2014]).

**Corollary 5.1.** *There exists a universal constant $C_0 > 0$ such that the following holds. Let $\mathcal{N}^* : \mathbb{R} \to \mathbb{R}$ be a depth-2 ReLU network of width $r$. Let $\varepsilon, \delta \in (0, 1)$ and let*

$$n \geq C_0 \cdot \frac{r \log(1/\varepsilon) + \log(1/\delta)}{\varepsilon} \, .$$

*Let $S$ be a size-$n$ binary classification dataset drawn from a distribution $\mathcal{D}$ and labeled according to $\mathcal{N}^*$. Consider GF on a depth-2 neural network $\mathcal{N}_{\boldsymbol{\theta}}$ w.r.t. the dataset $S$, and suppose that there exists time $t_0$ such that*

$$\mathcal{L}(\boldsymbol{\theta}(t_0)) < \frac{1}{n}.$$

*Then, GF converges in direction to $\boldsymbol{\theta}^*$ such that the network $\mathcal{N}_{\boldsymbol{\theta}^*}$ has at most $32r + 67$ linear regions, and with probability at least $1 - \delta$ over the sampling of $S$ we have*

$$\mathbb{P}_{x \sim \mathcal{D}} \left[\operatorname{sign}(\mathcal{N}_{\boldsymbol{\theta}^*}(x)) \neq \operatorname{sign}(\mathcal{N}^*(x))\right] \leq \varepsilon .$$

## Acknowledgments and Disclosure of Funding

Work done while GV was at the Weizmann Institute of Science. We thank Noam Razin and Gilad Yahudai for pointing out several relevant papers to discuss in the related work section.

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
