# A    Preliminaries on the Clarke subdifferential and the KKT conditions

Below we define the Clarke subdifferential, and review the definition of the KKT conditions for non-smooth optimization problems (cf. Lyu and Li [2019], Dutta et al. [2013]).

Let $f : \mathbb{R}^d \to \mathbb{R}$ be a locally Lipschitz function. The Clarke subdifferential [Clarke et al., 2008] at $\mathbf{x} \in \mathbb{R}^d$ is the convex set

$$\partial^\circ f(\mathbf{x}) := \operatorname{conv}\left\{ \lim_{i \to \infty} \nabla f(\mathbf{x}_i) \;\Big|\; \lim_{i \to \infty} \mathbf{x}_i = \mathbf{x}, \; f \text{ is differentiable at } \mathbf{x}_i \right\} .$$

If $f$ is continuously differentiable at $\mathbf{x}$ then $\partial^\circ f(\mathbf{x}) = \{\nabla f(\mathbf{x})\}$. For the Clarke subdifferential the chain rule holds as an inclusion rather than an equation. That is, for locally Lipschitz functions $z_1, \ldots, z_n : \mathbb{R}^d \to \mathbb{R}$ and $f : \mathbb{R}^n \to \mathbb{R}$, we have

$$\partial^\circ (f \circ \mathbf{z})(\mathbf{x}) \subseteq \operatorname{conv}\left\{ \sum_{i=1}^{n} \alpha_i \mathbf{h}_i : \boldsymbol{\alpha} \in \partial^\circ f(z_1(\mathbf{x}), \ldots, z_n(\mathbf{x})), \mathbf{h}_i \in \partial^\circ z_i(\mathbf{x}) \right\} .$$

Consider the following optimization problem

$$\min f(\mathbf{x}) \quad \text{s.t.} \quad \forall n \in [N] \; g_n(\mathbf{x}) \le 0 , \tag{7}$$

where $f, g_1, \ldots, g_n : \mathbb{R}^d \to \mathbb{R}$ are locally Lipschitz functions. We say that $\mathbf{x} \in \mathbb{R}^d$ is a *feasible point* of Problem (7) if $\mathbf{x}$ satisfies $g_n(\mathbf{x}) \le 0$ for all $n \in [N]$. We say that a feasible point $\mathbf{x}$ is a *KKT point* if there exists $\lambda_1, \ldots, \lambda_N \ge 0$ such that

1. $\mathbf{0} \in \partial^\circ f(\mathbf{x}) + \sum_{n \in [N]} \lambda_n \partial^\circ g_n(\mathbf{x})$;
2. For all $n \in [N]$ we have $\lambda_n g_n(\mathbf{x}) = 0$.

# B    Details on Example 1

We use here the notation $\Phi(\boldsymbol{\theta}; x) = \mathcal{N}_{\boldsymbol{\theta}}(x)$, and denote by $\sigma'(\cdot)$ a sub-gradient of $\sigma$, namely, $\sigma(z) = \mathbb{1}[z > 0]$ if $z \ne 0$ and $\sigma'(0) \in [0, 1]$ (the exact value in this case is not important here). For every $j \in \{1, 2\}$ we have

$$\nabla_{w_j} \mathcal{L}(\boldsymbol{\theta}) = \frac{1}{2} \sum_{i=1}^{2} \ell'(y_i \Phi(\boldsymbol{\theta}; x_i)) \cdot y_i \nabla_{w_j} \Phi(\boldsymbol{\theta}; x_i)$$

$$= \frac{1}{2} \sum_{i=1}^{2} \ell'(v_1 \sigma(w_1 x_i + b_1) + v_2 \sigma(w_2 x_i + b_2)) \cdot v_j \sigma'(w_j x_i + b_j) x_i .$$

Likewise,

$$\nabla_{v_j} \mathcal{L}(\boldsymbol{\theta}) = \frac{1}{2} \sum_{i=1}^{2} \ell'(y_i \Phi(\boldsymbol{\theta}; x_i)) \cdot y_i \nabla_{v_j} \Phi(\boldsymbol{\theta}; x_i)$$

$$= \frac{1}{2} \sum_{i=1}^{2} \ell'(v_1 \sigma(w_1 x_i + b_1) + v_2 \sigma(w_2 x_i + b_2)) \cdot \sigma(w_j x_i + b_j)$$

and

$$\nabla_{b_j} \mathcal{L}(\boldsymbol{\theta}) = \frac{1}{2} \sum_{i=1}^{2} \ell'(y_i \Phi(\boldsymbol{\theta}; x_i)) \cdot y_i \nabla_{b_j} \Phi(\boldsymbol{\theta}; x_i)$$

$$= \frac{1}{2} \sum_{i=1}^{2} \ell'(v_1 \sigma(w_1 x_i + b_1) + v_2 \sigma(w_2 x_i + b_2)) \cdot v_j \sigma'(w_j x_i + b_j) .$$

Note that if $w_2 = b_2 = v_2 = 0$ then we have $\nabla_{w_2} \mathcal{L}(\boldsymbol{\theta}) = \nabla_{v_2} \mathcal{L}(\boldsymbol{\theta}) = \nabla_{b_2} \mathcal{L}(\boldsymbol{\theta}) = 0$. Since these parameters are initialized at zero, then they remain zero throughout the training. Moreover, Suppose that $w_1 = 0$ and $b_1 = v_1 = \alpha$ for some $\alpha > 0$, and that $w_2 = b_2 = v_2 = 0$, then we have

$$-\frac{dw_1}{dt} = \nabla_{w_1} \mathcal{L}(\boldsymbol{\theta}) = \frac{1}{2} \sum_{i=1}^{2} \ell'(\alpha \sigma(\alpha)) \cdot \alpha \sigma'(\alpha) x_i = \frac{1}{2} \left( \ell'(\alpha^2) \cdot \alpha \cdot 4 + \ell'(\alpha^2) \cdot \alpha \cdot (-4) \right) = 0 ,$$

$$-\frac{dv_1}{dt} = \nabla_{v_1}\mathcal{L}(\boldsymbol{\theta}) = \frac{1}{2}\sum_{i=1}^{2}\ell'(\alpha\sigma(\alpha))\cdot\sigma(\alpha) = \frac{1}{2}\sum_{i=1}^{2}\ell'(\alpha^2)\cdot\alpha\,,$$

$$-\frac{db_1}{dt} = \nabla_{b_1}\mathcal{L}(\boldsymbol{\theta}) = \frac{1}{2}\sum_{i=1}^{2}\ell'(\alpha\sigma(\alpha))\cdot\alpha\sigma'(\alpha) = \frac{1}{2}\sum_{i=1}^{2}\ell'(\alpha^2)\cdot\alpha\,.$$

Hence, for every $t$ we have $w_1(t) = 0$ and $b_1(t) = v_1(t) = \alpha(t)$ where $\alpha(t) > 0$ is monotonically increasing.

As a result, the KKT point $\boldsymbol{\theta}^*$ is such that $w_2^* = b_2^* = v_2^* = w_1^* = 0$, and $b_1^* = v_1^* = \alpha^*$ for some $\alpha^* > 0$. Since $\boldsymbol{\theta}^*$ satisfies the KKT conditions of Problem (4), then we have

$$\alpha^* = b_1^* = \sum_{i=1}^{2}\lambda_i y_i \nabla_{b_1}\Phi(\boldsymbol{\theta}^*; x_i)\,,$$

where $\lambda_i \geq 0$ and $\lambda_i = 0$ if $y_i\Phi(\boldsymbol{\theta}^*; x_i) \neq 1$. Hence, there is $i$ such that $y_i\Phi(\boldsymbol{\theta}^*; x_i) = 1$. Thus, we have $1 = y_i\Phi(\boldsymbol{\theta}^*; x_i) = (\alpha^*)^2$ which implies $\alpha^* = 1$. Therefore $\boldsymbol{\theta}^* = \boldsymbol{\theta}(0)$.

## C  Proof of Theorem 3.1

Before we prove the theorem, we first state a few definitions that are specific for this appendix. Let $S = \{(x_i, y_i)\}_{i=1}^{n} \subseteq [-R, R] \times \{-1, 1\}$ be a dataset such that $x_1 < \ldots < x_n$ and let $I = \{i \in [n-1] : y_i \neq y_{i+1}\}$ where we denote the elements of $I$ using $i_1 < \ldots < i_r$, and $i_0 = -R, i_{r+1} = R$, where $r = |I|$. For all $j \in [r+1]$, define $I_j = (i_{j-1}, i_j + 1)$ which is the $j$-th interval where the instances in the data do not change their classification.[5] Given some function $\mathcal{L}$ and real number $\alpha$, we let $L_\alpha^+(\mathcal{L}) := \{\boldsymbol{\theta} : \mathcal{L}(\boldsymbol{\theta}) \geq \alpha\}$ denote the $\alpha$-superlevel set of $\mathcal{L}$.

Next, we state the following definitions, which establish sufficient conditions for our objective function to be well-behaved in the sense of having a strict direction of descent in a certain neighborhood.

**Definition C.1** (Separability). *Under Assumption 2.2, we say that $\boldsymbol{\theta}$ is separable from $S$ with positive constants $\gamma, m < M, q < Q$ if for all $j \in [r+1]$, there exist three neurons with weights and biases denoted by $w_i(I_j)$ and $b_i(I_j)$ for $i \in [3]$, and breakpoints $\beta_1(I_j) < \beta_2(I_j) < \beta_3(I_j) \in I_j$, which satisfy the following items:*

1. *$m \leq |w_i(I_j)|$ and $|b_i(I_j)| \leq M$ for all $i \in [3]$.*

2. *There exist four neurons, two with breakpoints in each of the intervals $(-\infty, 0), (0, \infty)$, that are distinct from the neurons in the previous item, are active on all the data instances and whose weights $w_i', b_i'$ satisfy $m \leq |w_i'|$ and $|b_i'| \leq M$ for all $i \in [4]$. Moreover, their breakpoints satisfy $|\beta_i| \leq Q$ for all $i \in [4]$.*

3. *$q \leq \beta_2(I_j) - \beta_1(I_j), \beta_3(I_j) - \beta_2(I_j) \leq Q$ for all $j \in [r+1]$.*

4. *$|\beta_1(I_{j+1}) - \beta_3(I_j)| \leq Q$ for all $j \in [r]$.*

5. *$x_{i_j+1} - x_{i_j} \geq \gamma$ for all $j \in [r]$.*

*If the triplet of neurons satisfying the above items in an interval is not distinct, we assume w.l.o.g. that $\beta_1(\cdot)$ and $\beta_3(\cdot)$ return the left-most and right-most breakpoints satisfying the above, respectively.*

The following definition is used to describe a neighborhood around the initialization point in which our separability assumption above holds.

**Definition C.2** ($\Delta$-hidden Neighborhood). *Given a network $\mathcal{N}(\cdot)$, weights $\boldsymbol{\theta}$ and a constant $\Delta \geq 0$, we define the $\Delta$-hidden Neighborhood of $\mathcal{N}$ at $\boldsymbol{\theta}$ as the set*

$$U_\Delta(\boldsymbol{\theta}) := \left\{\boldsymbol{\theta}' = [\mathbf{w}', \mathbf{b}', \mathbf{v}'] : \left\|(w_j, b_j) - (w_j', b_j')\right\|_2 \leq \Delta \quad \forall j \in [k], \quad \mathbf{v}' \in \mathbb{R}^k\right\}.$$

---

[5]We note that these intervals overlap and thus do contain instances that change classification with respect to the teacher network, but not with respect to the sample.

That is, the neighborhood of balls of radius $\Delta$ centered at each hidden neuron of $\boldsymbol{\theta}$ and where the output neuron weights are arbitrary.

Following the above definitions, the following auxiliary lemmas will be used in the proof of the theorem. The technical lemma below establishes that a certain binary matrix is invertible and provides a bound on the spectral norm of its inverse.

**Lemma C.1.** *Suppose that $A \in \{0,1\}^{d \times d}$, such that the first row of $A$ is all-ones, and each subsequent row $i$ is either $(1, \ldots, 1, 0, \ldots, 0)$ with $i-1$ leading ones or $(0, \ldots, 0, 1, \ldots, 1)$ with $i-1$ leading zeros. Then $A$ is invertible and we have $\left\| A^{-1} \right\|_{sp} \leq d$.*

*Proof.* The invertability of $A$ follows from the fact that the first row of $A$ is an all-ones vector, since we can use elementary row operations to change all subsequent rows to start with a '0' and end with a '1' if needed, resulting in an upper triangular matrix with all-ones on its main diagonal which is thus invertible. To bound $\left\| A^{-1} \right\|_{sp}$, let $I_d \in \{0,1\}^{d \times d}$ denote the identity matrix. We will use Gaussian elimination to compute the entries of $A^{-1}$. We first subtract the first row from all the other rows that do not have leading zeros and then multiply by the constant $-1$. Performing the same operation on $I_d$ results in a matrix $B$ whose rows are either standard unit vectors or the vector $(1, \ldots, 1, 0, 1, \ldots, 1)$. The resulting matrix after performing these operations on $A$ is and upper triangular matrix with ones in all of its diagonal and above the diagonal entries. Since it is readily seen that the inverse of such a matrix is a matrix with all zero entries except for the main diagonal which is all-ones and the first diagonal above it which comprises of all $-1$'s. Denote this matrix using $B'$, we have that the inverse of $A$ is given by $B \cdot B'$. The entries of $A^{-1}$ therefore must consist of dot products of a standard unit vector and vectors $(0, \ldots, 0, 1, -1, 0, \ldots, 0)$, or the vector $(1, \ldots, 1, 0, 1, \ldots, 1)$ and vectors $(0, \ldots, 0, 1, -1, 0, \ldots, 0)$. In both cases the dot product is an element of $\{-1, 0, 1\}$, and therefore we can bound $\left\| A^{-1} \right\|_{sp}$ by the Frobenius norm of $A^{-1}$ which is at most $d$. $\qquad \square$

The following key lemma establishes that when $\boldsymbol{\theta}$ is separable from $S$ then there exists a direction in weight space which strictly decreases our objective value.

**Lemma C.2.** *Under Assumption 2.2, suppose that $\boldsymbol{\theta}$ is separable from $S$ with constants $\gamma, m, M, q, Q$, and that $\mathcal{L}(\boldsymbol{\theta}) \geq \frac{1}{2n}$. Then*

$$\frac{1}{2} \left\| \nabla \mathcal{L}(\boldsymbol{\theta}) \right\|_2^2 \geq \frac{\gamma^2 q^2 m^6}{259200 n^4 Q^2 M^4}.$$

*Proof.* Since $\mathcal{L}(\boldsymbol{\theta}) > \frac{1}{2n}$, there must exist some $i \in [n]$ such that $\ell(y_i \Phi(\boldsymbol{\theta}; x_i)) > \frac{1}{2n}$. We now consider two possible cases, depending on the location of $x_i$ with respect to the breakpoints whose existence is guaranteed by Def. C.1, where will show the existence of a direction $\mathbf{u}$ which guarantees that $\mathcal{L}(\boldsymbol{\theta})$ is strictly decreasing.

- Suppose that $x_i \in I_\ell$ for some $\ell \in [r+1]$, such that $x_i \in (\beta_1(I_\ell), \beta_3(I_\ell))$. Assume without loss of generality that $x_i \in (\beta_2(I_\ell), \beta_3(I_\ell))$ (the proof is symmetric otherwise), and for ease of notation denote $\beta_2 := \beta_1(I_\ell), \beta_3 := \beta_2(I_\ell), \beta_4 := \beta_3(I_\ell)$. Then by Item 2 in the separability assumption, there exists a breakpoints $\beta_1 < \beta_2$ such that all data instances in $(\beta_2, \beta_4)$ have the same classification and the neuron with breakpoint at $\beta_1$ is active on all the data instances. Moreover, Item 2 also guarantees the existence of two breakpoints that are distinct from the previous ones, which we denote by $\beta_5, \beta_6$, where $\beta_5 < \beta_1 < 0$ and $\beta_6 > 0$ are active on all the data points. We will now show the existence of a depth-2 ReLU network which consists of six hidden neurons with weights $\mathbf{w} = (w_1, \ldots, w_6)$ and biases $\mathbf{b} = (b_1, \ldots, b_6)$ (corresponding to the breakpoints $\beta_1, \ldots, \beta_6$ defined above) which computes the piece-wise linear function $f : [\beta_1, \beta_6) \to \mathbb{R}$ given by

$$f(x) := \begin{cases} 0 & x \in [\beta_1, \beta_2] \\ \frac{1}{\beta_3 - \beta_2} x - \frac{\beta_2}{\beta_3 - \beta_2} & x \in (\beta_2, \beta_3) \\ \frac{1}{\beta_3 - \beta_4} x - \frac{\beta_4}{\beta_3 - \beta_4} & x \in [\beta_3, \beta_4] \\ 0 & x \in (\beta_4, \beta_6) \end{cases}.$$

  The intuition behind the approximation is that we can use the first four neurons to approximate the slopes of the function $f$, and the last two remaining neurons to simulate a bias term

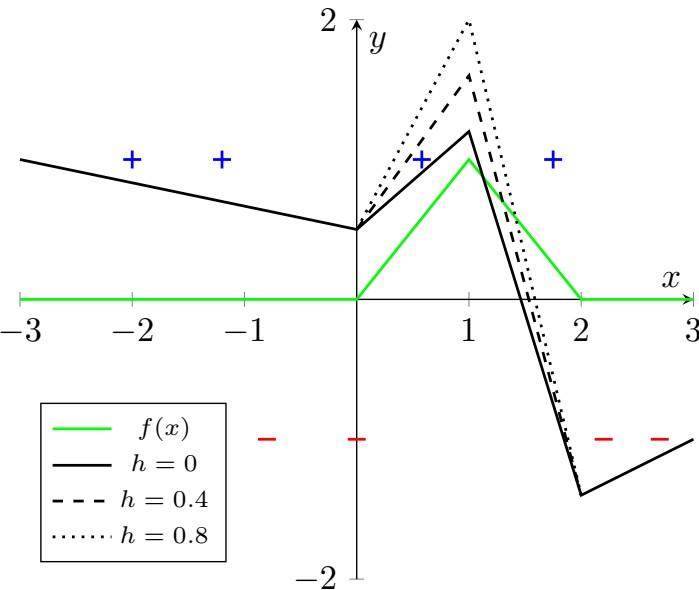

Figure 1: The plots of $f(x)$ (green) and $\Phi(\boldsymbol{\theta}; x) + h \cdot f(x)$ (black) for various values of $h$. Moving $\boldsymbol{\theta}$ in the direction of $\mathbf{u}$ which computes $f(\cdot)$ strictly decreases the loss over the positively-labeled instances in the interval $(0, 2)$ without affecting the rest of the dataset. Best viewed in color.

which would shift the function approximated by the network to overlap $f$ in the relevant domain of approximation (see Figure 1 for an illustration).

More formally, define $W := \text{diag}(w_1, \ldots, w_4) \in \mathbb{R}^{4 \times 4}$, define the masking matrix $A \in \mathbb{R}^{4 \times 4}$ with entries $a_{j,j'} = \mathbb{1}\{j\text{-th neuron is active on the interval starting with } \beta_{j'}\}$ and let

$$\mathbf{d} = \left(0, \frac{1}{\beta_3 - \beta_2}, \frac{1}{\beta_3 - \beta_4}, 0\right).$$

Thus, to match the slopes computed by the a depth-2 ReLU network with weights $\mathbf{w}, \mathbf{b}$ to those of $f$, we first want the output neuron's weights $\mathbf{v} = [\mathbf{v}_1, \mathbf{v}_2] \in \mathbb{R}^4 \times \mathbb{R}^2$ to satisfy the equality $A^\top W \mathbf{v}_1 = \mathbf{d}$. To this end, we have by Lemma C.1 that $A$ is invertible and $\left\|A^{-\top}\right\|_{\text{sp}} \leq 4$. It then follows that $\mathbf{v}_1 = W^{-1} A^{-\top} \mathbf{d}$, which entails

$$\|\mathbf{v}_1\|_2 \leq \left\|W^{-1}\right\|_{\text{sp}} \left\|A^{-\top}\right\|_{\text{sp}} \|\mathbf{d}\|_2 \cdot \leq \frac{8\sqrt{2}}{qm}, \tag{8}$$

where we used the separability assumption, implying that $\|\mathbf{d}\|_2 \leq \sqrt{2} q^{-1}$ due to Item 3 which guarantees that $\beta_3 - \beta_2, \beta_4 - \beta_3 \geq q$, and the lower bound assumption $|w_i| \geq m$ for all $i \in [4]$ which holds by Items 1 and 2. Next, we use the two neurons with breakpoints at $\beta_5, \beta_6$ to shift the network by a constant so that it overlaps with $f$ on the interval $[-R, R]$. To perform this shift, we first compute the magnitude by which we wish to shift which is given by the expression

$$b_0 := -\sum_{j=1}^{4} v_j b_j \mathbb{1}\{w_j x + b_j > 0 \ \forall x \in (\beta_{j'}, \beta_{j'+1})\}.$$

Letting

$$P := \begin{pmatrix} b_5 & b_6 \\ w_5 & w_6 \end{pmatrix},$$

we have that the neurons with breakpoints at $\beta_5, \beta_6$ compute a function which equals $b_0$ on the interval $[\beta_5, \beta_6]$ when the equality $P \cdot \mathbf{v}_2 = (b_0, 0)^\top$ is satisfied. We will now compute $P^{-1}$ and show that it is well-defined. The inverse of a $2 \times 2$ matrix is given by

$$P^{-1} = \frac{1}{b_5 w_6 - b_6 w_5} \begin{pmatrix} w_6 & -b_6 \\ -w_5 & b_5 \end{pmatrix} = \frac{1}{\beta_6 - \beta_5} \begin{pmatrix} \frac{1}{w_5} & -\frac{b_6}{w_5 w_6} \\ -\frac{1}{w_6} & \frac{b_5}{w_5 w_6} \end{pmatrix}.$$

Using the above, we can upper bound the spectral norm of $P^{-1}$ by upper bounding $1/(\beta_6 - \beta_5)$ with $\frac{1}{2R} \leq \frac{1}{2}$ since $\beta_5, \beta_6$ are outside the interval $[-R, R]$ at opposite sides and $R \geq 1$ by Assumption 2.2, and by upper bounding the spectral norm of the matrix with its Frobenius norm by using Item 2, to obtain

$$\left\| P^{-1} \right\|_{\text{sp}} \leq \frac{M}{m^2}.$$

Similarly, we derive an upper bound on $|b_0|$ using Cauchy-Schwartz and Items 1 and 2 to obtain

$$|b_0| \leq \|\mathbf{v}_1\|_2 \|\mathbf{b}\|_2 \leq 2M \|\mathbf{v}_1\|_2.$$

With the above, we can bound the norm of $\mathbf{v}_2$ as follows

$$\|\mathbf{v}_2\|_2 = \left\| P^{-1} \cdot (b_0, 0)^\top \right\|_{\text{sp}} \leq \frac{2M^2}{m^2} \|\mathbf{v}_1\|_2. \tag{9}$$

We now define $\mathbf{u}$ as the all-zero vector, except for the six output neuron entries corresponding to the neurons with breakpoints $\beta_1, \ldots, \beta_6$, where the coordinates of $\mathbf{u}$ take the values $y_i v_1, \ldots, y_i v_6$. Note that this entails

$$\|\mathbf{u}\|_2 = \sqrt{\|\mathbf{v}_1\|_2^2 + \|\mathbf{v}_2\|_2^2} \leq \|\mathbf{v}_1\|_2 \sqrt{1 + \frac{4M^4}{m^4}} \leq \frac{8}{qm} \sqrt{2 + \frac{8M^4}{m^4}} \leq 8\sqrt{10} \frac{M^2}{qm^3}, \tag{10}$$

where we have used Eq. (8) and the fact that $1 < M/m$. Next, we have for all $j \in [n]$ and $h > 0$ that

$$y_j \Phi\left(\boldsymbol{\theta} + \frac{h}{\|\mathbf{u}\|} \mathbf{u}; x_j\right) = y_j \left(\Phi(\boldsymbol{\theta}; x_j) + \frac{h}{\|\mathbf{u}\|} y_i f(x_j)\right).$$

Observe that all data points satisfy $x_j \in [\beta_5, \beta_6]$, and are therefore unaffected by the value $\Phi(\cdot, x)$ attains for $x$'s outside of this interval. Additionally, $f(x) = 0$ for all $x \in [\beta_1, \beta_2] \cup (\beta_4, \infty)$, which also keeps $\Phi(\cdot, x)$ unaffected by moving in the direction of $\mathbf{u}$. Moreover, since the sign of data instances in $x_j \in (\beta_2, \beta_4]$ is always $y_i$, we have that

$$y_j \Phi\left(\boldsymbol{\theta} + \frac{h}{\|\mathbf{u}\|} \mathbf{u}; x_j\right) = y_j \Phi(\boldsymbol{\theta}; x_j) + \frac{h}{\|\mathbf{u}\|} f(x_j) \geq y_j \Phi(\boldsymbol{\theta}; x_j).$$

Lastly, for $x_i$ we have that $x_i \in [\beta_3, \beta_4]$, and that $x_i$ is at distance at least $\gamma$ from the boundary by Item 5 in our separability assumption. By Item 3, this implies that $f(x_i)$ is at least $\frac{\gamma}{\beta_4 - \beta_3} \geq \frac{\gamma}{Q}$, which with the above equation and our bound from Eq. (10) yields

$$y_i \Phi\left(\boldsymbol{\theta} + \frac{h}{\|\mathbf{u}\|} \mathbf{u}; x_i\right) \geq y_i \Phi(\boldsymbol{\theta}; x_i) + h \frac{\gamma}{Q \|\mathbf{u}\|} \geq y_i \Phi(\boldsymbol{\theta}; x_i) + h \frac{\gamma q m^3}{8\sqrt{10} Q M^2}. \tag{11}$$

- Suppose that $x_i \in I_\ell$ for some $\ell \in \{1, \ldots, r+1\}$, such that $x_i \notin (\beta_1(I_\ell), \beta_3(I_\ell))$. Assume without loss of generality that $x_i \leq \beta_4 := \beta_1(I_\ell)$ (the proof is symmetric otherwise). Then by our separability assumption, there exist $\beta_2 := \beta_1(I_{\ell-1}), \beta_3 := \beta_3(I_{\ell-1}), \beta_5 := \beta_3(I_\ell)$ and $\beta_1$ whose neuron is active on all the data points where $\beta_1 \leq \beta_2$. Note that by the definition of $\beta_i(\cdot)$, it must hold that $\beta_4$ is the smallest element in the interval $I_\ell$ which satisfies Items 1 and 3 (since otherwise we would have that $x_i \in (\beta_3, \beta_5) \subseteq I_\ell$, which is handled in the previous case), and therefore $\beta_3 \notin I_\ell$, implying that $x_i \in (\beta_3, \beta_4)$. Moreover, we note that we may assume that $\ell > 1$, since otherwise we can take the smallest two breakpoints that are active on all the data along with $\beta_4$ which reduces us to the previous case. We thus denote the largest data instance in $I_{\ell-1}$ as $x_{i-1}$ which implies $x_{i-1} < x_i$. Lastly, Item 2 also guarantees the existence of two breakpoints that are distinct from the previous ones, which we denote by $\beta_6, \beta_7$, where $\beta_6 < \beta_1 < 0$ and $\beta_7 > 0$ are active on all the data points.

Following a similar approach as in the previous case, we will now show the existence of a depth-2 ReLU network which consists of seven hidden neurons with weights $\mathbf{w} = (w_1, \ldots, w_7)$ and biases $\mathbf{b} = (b_1, \ldots, b_7)$ and computes the piece-wise linear function

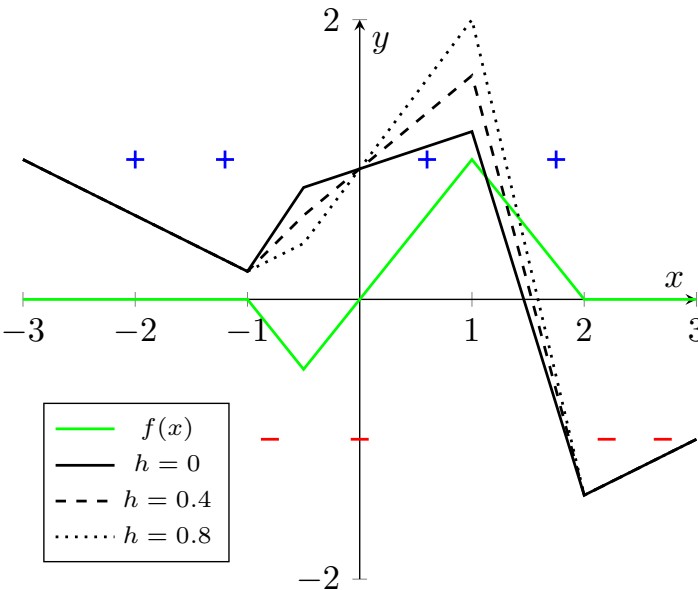

Figure 2: The plots of $f(x)$ (green) and $\Phi(\boldsymbol{\theta}; x) + h \cdot f(x)$ (black) for various values of $h$. Moving $\boldsymbol{\theta}$ in the direction of $\mathbf{u}$ which computes $f(\cdot)$ strictly decreases the loss over the positively-labeled instances in the interval $(0, 2)$ without degrading the loss over the rest of the dataset. Unlike the previous simpler case, since $\boldsymbol{\theta}$ has no breakpoints between the negative instance at $x = 0$ and the positive instance at $x = 0.6$, we use a function $f(x)$ which pivots around $x = 0$ to prevent the prediction over the negatively-labeled instances in the interval $[-1, 0]$ from increasing. Best viewed in color.

$f : [\beta_1, \beta_7) \to \mathbb{R}$ given by

$$
f(x) := \begin{cases}
0 & x \in [\beta_1, \beta_2] \\
\frac{1}{\beta_2 - \beta_3} x - \frac{\beta_2}{\beta_2 - \beta_3} & x \in (\beta_2, \beta_3) \\
\frac{1}{x_{i-1} - \beta_3} x - \frac{x_{i-1}}{x_{i-1} - \beta_3} & x \in [\beta_3, \beta_4] \\
\frac{\beta_4 - x_{i-1}}{x_{i-1} - \beta_3} \left( \frac{1}{\beta_4 - \beta_5} x - \frac{\beta_5}{\beta_4 - \beta_5} \right) & x \in (\beta_4, \beta_5) \\
0 & x \in [\beta_5, \beta_7)
\end{cases} ,
$$

where the first five neurons are used to compute the slopes of $f$ and the remaining last two neurons are used to simulate a bias term to shift the network to accord with $f$ in its domain (see Figure 2 for an illustration).

Define $W := \mathrm{diag}(w_1, \ldots, w_5) \in \mathbb{R}^{5 \times 5}$, define the masking matrix $A \in \mathbb{R}^{5 \times 5}$ with entries $a_{j,j'} = \mathbb{1}\{j\text{-th neuron is active on the interval starting with } \beta_{j'}\}$ and let

$$
\mathbf{d} = \left( 0, \frac{1}{\beta_2 - \beta_3}, \frac{1}{x_{i-1} - \beta_3}, \frac{\beta_4 - x_{i-1}}{x_{i-1} - \beta_3} \cdot \frac{1}{\beta_4 - \beta_5}, 0 \right).
$$

Thus, to match the slopes computed by the a depth-2 ReLU network with weights $\mathbf{w}, \mathbf{b}$ to those of $f$, we first want the output neuron's weights $\mathbf{v} = [\mathbf{v}_1, \mathbf{v}_2] \in \mathbb{R}^5 \times \mathbb{R}^2$ to satisfy the equality $A^\top W \mathbf{v}_1 = \mathbf{d}$. To this end, we have by Lemma C.1 that $A$ is invertible and $\left\| A^{-\top} \right\|_{\mathrm{sp}} \le 5$. It then follows that $\mathbf{v}_1 = W^{-1} A^{-\top} \mathbf{d}$, which entails

$$
\| \mathbf{v}_1 \|_2 \le \left\| W^{-1} \right\|_{\mathrm{sp}} \left\| A^{-\top} \right\|_{\mathrm{sp}} \| \mathbf{d} \|_2 \le 5 \left\| W^{-1} \right\|_{\mathrm{sp}} \| \mathbf{d} \|_2 .
$$

To upper bound the above, we first bound $\|\mathbf{d}\|_2$. By our separability assumption we have

$$\|\mathbf{d}\|_2 \le \sqrt{\frac{1}{q^2} + \frac{1}{(x_{i-1} - \beta_3)^2} + \frac{4Q^2}{q^2(x_{i-1} - \beta_3)^2}}$$

$$= \sqrt{\frac{q^2 + (x_{i-1} - \beta_3)^2 + 4Q^2}{q^2(x_{i-1} - \beta_3)^2}} \le \frac{1}{q(x_{i-1} - \beta_3)}\sqrt{8Q^2 + q^2}$$

where we used Item 3 to upper bound the denominators, and Item 4 which entails $\beta_4 - x_{i-1}, x_{i-1} - \beta_3 \le \beta_4 - \beta_3 \le Q$. By Item 1, we have $|w_i| \ge m$ for all $i \in [5]$, and thus we obtain

$$\|\mathbf{v}_1\|_2 \le \frac{5}{mq(x_{i-1} - \beta_3)}\sqrt{40Q^2 + 5q^2} \le \frac{15\sqrt{5}Q}{mq(x_{i-1} - \beta_3)}. \tag{12}$$

Now, similarly to the previous case, we wish to shift the function computed by the depth-2 ReLU network by a constant. This is done in the exact same manner as in the previous case, where we shift it by a magnitude given by

$$b_0 = -\sum_{j=1}^{5} v_j b_j \mathbb{1}\left\{w_j x + b_j > 0 \ \ \forall x \in (\beta_{j'}, \beta_{j'+1})\right\}.$$

Bounding $|b_0|$ using its above definition, Cauchy-Schwartz and Item 1 in our separability assumption, we obtain

$$|b_0| \le \|\mathbf{v}_1\|_2 \|\mathbf{b}\|_2 \le \sqrt{5}M \|\mathbf{v}_1\|_2.$$

From the above and Eq. (9), we can shift the network by the desired magnitude using a vector $\mathbf{v}_2 = (v_6, v_7)$ satisfying

$$\|\mathbf{v}_2\|_2 \le \frac{\sqrt{5}M^2}{m^2} \|\mathbf{v}_1\|_2.$$

We now define $\mathbf{u}$ as the all-zero vector, except for the output neuron entries corresponding to the neurons with breakpoints $\beta_1, \ldots, \beta_7$, where the coordinates of $\mathbf{u}$ take the values $y_i v_1, \ldots, y_i v_7$. Note that this entails

$$\|\mathbf{u}\|_2 = \sqrt{\|\mathbf{v}_1\|_2^2 + \|\mathbf{v}_2\|_2^2} \le \|\mathbf{v}_1\|_2 \sqrt{1 + \frac{5M^4}{m^4}} \le \frac{15\sqrt{30}QM^2}{m^3 q(x_{i-1} - \beta_3)} \le \frac{90QM^2}{m^3 q(x_{i-1} - \beta_3)}, \tag{13}$$

where we used Eq. (12) and the fact that $1 < M/m$. We therefore have for all $j \in [n]$ and $h > 0$ that

$$y_j \Phi\left(\boldsymbol{\theta} + \frac{h}{\|\mathbf{u}\|}\mathbf{u}; x_j\right) = y_j(\Phi(\boldsymbol{\theta}; x_j) + \frac{h}{\|\mathbf{u}\|} y_i f(x_j)).$$

Observe that all data points satisfy $x_j \in [\beta_6, \beta_7]$, and are therefore unaffected by the value $\Phi(\cdot, x)$ attains for $x$'s outside of this interval. Additionally, $f(x) = 0$ for all $x \in [\beta_1, \beta_2] \cup [\beta_5, \infty)$, which also keeps $\Phi(\cdot, x)$ unaffected by moving in the direction of $\mathbf{u}$. In the interval $(\beta_2, x_{i-1})$, the sign of $\Phi(\cdot, x)$ is $-y_i$, and in the interval $(x_{i-1}, \beta_5)$ its sign changes to $y_i$. For this reason, similarly to the previous case, we have that

$$y_j \Phi\left(\boldsymbol{\theta} + \frac{h}{\|\mathbf{u}\|}\mathbf{u}; x_j\right) = y_j \Phi(\boldsymbol{\theta}; x_j) + \frac{h}{\|\mathbf{u}\|}f(x_j) \ge y_j \Phi(\boldsymbol{\theta}; x_j), \tag{14}$$

for all $x_j \in (\beta_2, \beta_5)$. Lastly, for $x_i$ we have that $x_i \in [\beta_3, \beta_4]$, and that $x_i$ is at distance at least $\gamma$ from $x_{i-1}$ by Item 5 in our separability assumption. This implies that

$$f(x_i) = \frac{x_i}{x_{i-1} - \beta_3} - \frac{x_{i-1}}{x_{i-1} - \beta_3} \ge \frac{\gamma}{x_{i-1} - \beta_3},$$

and therefore $hf(x_i) \ge \frac{\gamma}{x_{i-1} - \beta_3} h$, which with Eqs. (13,14) implies that

$$y_i \Phi\left(\boldsymbol{\theta} + \frac{h}{\|\mathbf{u}\|}\mathbf{u}; x_i\right) \ge y_i \Phi(\boldsymbol{\theta}; x_i) + h\frac{\gamma}{\|\mathbf{u}\|(x_{i-1} - \beta_3)} \ge y_i \Phi(\boldsymbol{\theta}; x_i) + h\frac{\gamma q m^3}{90QM^2}, \tag{15}$$

which is a weaker lower bound than the one derived in Eq. (11), and thus always holds if Eq. (11) is satisfied.

We now turn to lower bound the norm of the gradient by analyzing the directional derivative of $\mathcal{L}(\cdot)$ in the direction of the vector $\mathbf{u}$ defined by the above two cases. To this end, denote $\bar{\mathbf{u}} = \frac{\mathbf{u}}{\|\mathbf{u}\|}$ and compute

$$\|\nabla\mathcal{L}(\boldsymbol{\theta})\|_2 \geq |\langle\nabla\mathcal{L}(\boldsymbol{\theta}), \bar{\mathbf{u}}\rangle| = \left|\lim_{h\to 0}\frac{1}{hn}\sum_{j=1}^{n}\left(\ell(y_j\Phi(\boldsymbol{\theta} + h\bar{\mathbf{u}}; x_j)) - \ell(y_j\Phi(\boldsymbol{\theta}; x_j))\right)\right|$$

$$= \lim_{h\to 0}\frac{1}{hn}\sum_{j=1}^{n}\left(\ell(y_j\Phi(\boldsymbol{\theta}; x_j)) - \ell(y_j\Phi(\boldsymbol{\theta} + h\bar{\mathbf{u}}; x_j))\right)$$

$$\geq \lim_{h\to 0}\frac{1}{hn}\left(\ell(y_i\Phi(\boldsymbol{\theta}; x_i)) - \ell(y_i\Phi(\boldsymbol{\theta} + h\bar{\mathbf{u}}; x_i))\right)$$

$$\geq \lim_{h\to 0}\frac{1}{hn}\left(\ell(y_i\Phi(\boldsymbol{\theta}; x_i)) - \ell\left(y_i\Phi(\boldsymbol{\theta}; x_i) + h\frac{\gamma qm^3}{90QM^2}\right)\right)$$

$$= -\frac{\gamma qm^3}{90nQM^2}\ell'(y_i\Phi(\boldsymbol{\theta}; x_i)). \tag{16}$$

In the above, the first inequality is by Cauchy-Schwartz; the second equality is due to Eq. (14), which guarantees that each summand is non-positive; the second inequality is another application of Eq. (14) which guarantees that we're omitting only non-negative terms; and the last inequality is due to Eq. (15). We now lower bound the above expression depending on whether $\ell(\cdot)$ is the exponential or the logistic loss.

First assume that $\ell(\cdot)$ is the exponential loss. Then we have that $-\ell'(x) = \ell(x)$ for all $x \in \mathbb{R}$, and therefore we can directly use the inequality $\ell(y_i\Phi(\boldsymbol{\theta}; x_i)) > 1/2n$.

In the case where $\ell(\cdot)$ is the logistic loss, we have that

$$-\ell'(x) = \frac{1}{1 + \exp(x)} = 1 - \frac{1}{1 + \exp(-x)}.$$

By the inequality $\ell(y_i\Phi(\boldsymbol{\theta}; x_i)) > 1/2n$, we have $1 + \exp(-y_i\Phi(\boldsymbol{\theta}; x_i)) > \exp(1/2n)$, implying that

$$-\ell'(y_i\Phi(\boldsymbol{\theta}; x_i)) > 1 - \exp\left(-\frac{1}{2n}\right) \geq \frac{1}{4n},$$

where we used the inequality $\exp(-x) \leq 1 - 0.5x$ which holds for all $x \in [0, 1]$. Combining both loss cases and Eq. (16), we arrived at

$$\|\nabla\mathcal{L}(\boldsymbol{\theta})\|_2 \geq \frac{\gamma qm^3}{360n^2QM^2}.$$

Squaring the above and dividing by 2, the lemma follows. $\qquad\square$

The following proposition establishes the separability (Def. C.1) of a neighborhood in weight space around our initialization point (Def. C.2) from the dataset $S$.

**Proposition C.1** (Bounded Gradient with High Probability). *Under Assumptions 2.1 and 2.2, given any $\delta \in (0, 1)$, suppose that the following hold*

$$k \geq 6144 \cdot \frac{R^4\log\left(\frac{24r}{\delta}\right)}{\rho} \quad and \quad \Delta = \frac{\delta\rho\sigma_h}{24nkCR^3}. \tag{17}$$

*Then with probability at least $1 - \delta$, for all $\boldsymbol{\theta} \in U_\Delta(\boldsymbol{\theta}(0)) \cap L_{1/2n}^+(\mathcal{L})$, we have that*

$$\frac{1}{2}\|\nabla\mathcal{L}(\boldsymbol{\theta})\|_2^2 \geq 3 \cdot 10^{-11}\frac{\delta^2\rho^2}{n^6r^2C^2R^8}\sigma_h^2.$$

*Proof.* To prove the proposition, we will show that for all $\boldsymbol{\theta} \in U_\Delta(\boldsymbol{\theta}(0)) \cap L_{1/2n}^+(\mathcal{L})$, $\boldsymbol{\theta}$ is separable from $S$ with high probability. By Lemma C.2, this would imply the proposition. We will begin with proving items 1-4 jointly, and then we will show item 5 separately.

1-4. Under Assumption 2.1, suppose we are given some $a \in [0, R]$ and $\boldsymbol{\xi} := (\xi_1, \xi_2) \in \{-1, 1\}^2$, and sufficiently small $\varepsilon > 0$ such that

$$\varepsilon \leq \frac{512R^4 \log\left(\frac{24r}{\delta}\right)}{k} \leq \frac{\rho}{6} \leq \frac{R}{6}, \tag{18}$$

where the first inequality is by assumption on $\varepsilon$, the second inequality is by the lower bound on $k$ in Eq. (17), and the last inequality follows from $\rho \leq \frac{2R}{r+1} \leq R$ since $\rho$ must be smaller than the average length of an interval and since we assume $r \geq 1$. We now consider the event denoted by $E_{a,\varepsilon,\boldsymbol{\xi}}$ where the weights $w_i, b_i$ of the $i$-th neuron satisfy

$$w_i \in \left(\xi_1 \frac{\sigma_{\mathrm{h}}}{R}, \xi_1 \frac{2\sigma_{\mathrm{h}}}{R}\right) \quad \text{and} \quad -\frac{b_i}{w_i} \in (\xi_2 a, \xi_2(a + \varepsilon)), \tag{19}$$

Since such an event is symmetric about 0, it is unaffected by the signs of $\boldsymbol{\xi}$. We can therefore assume without loss of generality that both intervals in Eq. (19) are contained in the positive real line and omit $\boldsymbol{\xi}$ from our notation. Under this assumption, the probability of $E_{a,\varepsilon}$ can be given in terms of Owen's T function which is defined by

$$T(h, a) := \frac{1}{2\pi} \int_0^a \frac{\exp\left(-\frac{1}{2}h^2(1 + x^2)\right)}{1 + x^2} dx$$

(see Owen [1956]), yielding

$$\mathbb{P}[E_{a,\varepsilon}] = T\left(\frac{1}{R}, a + \varepsilon\right) - T\left(\frac{1}{R}, a\right) - \left(T\left(\frac{2}{R}, a + \varepsilon\right) - T\left(\frac{2}{R}, a\right)\right).$$

Using the definition of $T(\cdot, \cdot)$, the above can be simplified to

$$\begin{aligned}
\mathbb{P}[E_{a,\varepsilon}] &= \frac{1}{2\pi} \int_a^{a+\varepsilon} \frac{\exp\left(-\frac{1}{2R^2}(1 + x^2)\right)\left(1 - \exp\left(-\frac{3}{2R^2}(1 + x^2)\right)\right)}{1 + x^2} dx \\
&\geq \frac{1}{2\pi}\left(1 - \exp\left(-\frac{3}{2R^2}\right)\right) \int_a^{a+\varepsilon} \frac{\exp\left(-\frac{1}{2R^2}(1 + x^2)\right)}{1 + x^2} dx \\
&\geq \frac{3}{8\pi R^2} \int_a^{a+\varepsilon} \frac{\exp\left(-\frac{1}{2R^2}(1 + x^2)\right)}{1 + x^2} dx \\
&\geq \frac{3}{8\pi R^2} \int_a^{a+\varepsilon} \frac{\exp\left(-\frac{1}{2R^2}(1 + (R + \varepsilon)^2)\right)}{1 + (R + \varepsilon)^2} dx \geq \frac{\varepsilon}{512R^4}. \tag{20}
\end{aligned}$$

In the above, the second inequality follows from the inequality $1 - \exp(-x) \geq 0.5x$ which holds for all $x \in [0, 1.5]$ and from the fact that $1 \leq R$; the third inequality follows from the fact that the integrand is a monotonically decreasing function and $|a| \leq R$; and the last inequality follows from $\varepsilon \leq R$ which is implied by Eq. (18) and allows us to lower bound the numerator of the integrand by $\exp(-2.5)$ and upper bound the denominator by $5R^2$, and the fact that $3/(40\pi \exp(2.5)) \geq 1/512$.

Next, given some interval $I_j := (x_{i_j}, x_{i_{j+1}+1})$, $j \in [r + 1]$, where the classification does not change signs on the data, we consider the three sub-intervals given by

$$\begin{aligned}
I_{j_1} &:= (x_{i_j}, x_{i_j} + \varepsilon), \\
I_{j_2} &:= \left(\frac{x_{i_j} + x_{i_{j+1}+1}}{2} - \frac{\varepsilon}{2}, \frac{x_{i_j} + x_{i_{j+1}+1}}{2} + \frac{\varepsilon}{2}\right), \\
I_{j_3} &:= \left(x_{i_{j+1}+1} - \varepsilon, x_{i_{j+1}+1}\right).
\end{aligned}$$

We remark that due to Eq. (18), the above sub-intervals are all disjoint and the distance between the intervals is positive. We now wish to show that Items 1 and 3 hold. We have from Eq. (20) that the probability that a given sub-interval of length $\varepsilon$ contains no breakpoint is at most

$$(1 - \mathbb{P}[E_{a,\varepsilon}])^k \leq \left(1 - \frac{\log\left(\frac{24r}{\delta}\right)}{k}\right)^k \leq \exp\left(-\log\left(\frac{24r}{\delta}\right)\right) = \frac{\delta}{24r},$$

where we used the inequality $(1 - x/y)^y \leq \exp(-x)$ which holds for all $x, y > 0$. There are exactly $3 \cdot (r+1) \leq 6r$ sub-intervals, therefore by a union bound we have that Items 1 and 3 hold for some positive $q, Q, m, M$ with probability at least $1 - \frac{\delta}{4}$.

Next, we show Item 2. By Eq. (20), we have

$$\mathbb{P}\left[E_{R,R/6}\right] \geq \frac{1}{3072R^3}.$$

Thus, the probability of initializing a neuron with breakpoint in $(R, \frac{7}{6}R)$ which is active on all the data points is at least $\frac{1}{6144R^3}$, since there's an independent $0.5$ probability that it has the correct orientation. This entails that the probability of initializing at most one neuron which is active on all the data points and has a breakpoint in $(R, \frac{7}{6}R)$ is upper bounded by

$$\left(1 - \mathbb{P}\left[E_{R,R/6}\right]\right)^k + k\left(1 - \mathbb{P}\left[E_{R,R/6}\right]\right)^{k-1}\mathbb{P}\left[E_{R,R/6}\right] \leq 2\left(1 - \frac{\mathbb{P}\left[E_{R,R/6}\right]}{2}\right)^k$$

$$\leq 2\left(1 - \frac{1}{6144R^3}\right)^k \leq 2\exp\left(-\frac{k}{6144R^3}\right)$$

$$\leq 2\exp\left(-\frac{R\log(24r/\delta)}{\rho}\right) \leq \frac{2\delta}{24r} \leq \frac{\delta}{12}.$$

In the above, the first inequality follows from the inequality $(1-x)^k + k(1-x)^{k-1}x \leq 2(1-x/2)^k$ which holds for any natural $k$ and all $x \in [0,1]$,[6] the third inequality follows from $(1-1/x)^x \leq \exp(-1)$ for all $x > 0$, the fourth inequality follows from our lower bound on $k$ in Eq. (17), and the penultimate inequality holds due to Eq. (18) which entails $\rho \leq R$. Therefore, by the above and a union bound on the symmetric event where two neurons are initialized in $(-\frac{7}{6}R, -R)$, we have that Item 2 holds for some $m, M$ with probability at least $1 - \frac{\delta}{6} \geq 1 - \frac{\delta}{4}$.

We will now derive explicit bounds on the constants $m, M, q, Q$, and in addition we will show that Item 4 holds. Applying a union bound on the two previous cases, we have that Items 1-3 hold with probability at least $1 - \delta/2$. In such a case, we get an explicit lower bound on $q$ as follows

$$q \geq \frac{|I_j| - 3\varepsilon}{2} \geq \frac{\rho - 3\varepsilon}{2} \geq \frac{\rho}{4},$$

where in the second inequality we used the fact that $|I_j| \geq \rho$ for all $j \in [r+1]$ which holds by the definition of $\rho$ and in the last inequality we used Eq. (18). To bound $Q$ in Item 4, we first argue that under the realization of $E_{R,R/6}$, the four neurons that are active on all the data points have a breakpoint with absolute value at most $\frac{7}{6}R \leq 2R$. To upper bound $Q$ in Item 3, observe that under the realization of the previous events we have that

$$\beta_{i+1}(I_j) - \beta_i(I_j) \leq \frac{|I_j| + \varepsilon}{2} \leq R + \frac{\varepsilon - \rho}{2} \leq R,$$

for all $j \in [r+1]$ and $i \in [2]$, which follows from $|I_j| \leq 2R - \rho$ since $r \geq 1$ and $I_j \subseteq [-R, R]$ (i.e. there exists at least one interval other than $I_j$ which has length at least $\rho$), and from the inequality $\varepsilon \leq \rho/6$ which holds by Eq. (18). To upper bound $Q$ in Item 4, we bound the term $|\beta_1(I_{j+1}) - \beta_3(I_j)|$. Observe that under the realization of the above events we have that $\beta_1(I_{j+1}) \in (x, x+\varepsilon)$ and $\beta_3(I_j) \in (x' - \varepsilon, x')$ where $x < x'$ are the largest and smallest data instances in $I_j, I_{j+1}$, respectively. We therefore have

$$|\beta_1(I_{j+1}) - \beta_3(I_j)| \leq \max\{|x - x'|, |x - x' + 2\varepsilon|\} \leq |x - x'| + 2\varepsilon \leq \frac{7}{3}R,$$

where the second inequality follows from the triangle inequality and the last inequality follows from the fact that $x_i \in [-R, R]$ for all $i \in [n]$ and from Eq. (18). Turning to bound

---

[6]To show this inequality holds, consider $k$ i.i.d. random variables $X_j \sim U([0,1])$. Then the left-hand side equals $\mathbb{P}[|\{x_j : x_j \in [0,x]\}| \leq 1]$. The occurrence of the complement of this event is implied if $x_i \in [0, x/2]$ and $x_{i'} \in [x/2, x]$ hold for some $i \neq i'$, therefore to upper bound the left-hand side it suffices to upper bound the complement of the event where $x_i \in [0, x/2]$ and $x_{i'} \in [x/2, x]$ hold for some $i \neq i'$. This in turn follows from applying a union bound on $\mathbb{P}[|\{x_j : x_j \in [0, x/2]\}| = 0]$ and $\mathbb{P}[|\{x_j : x_j \in [x/2, x]\}| = 0]$.

$w_i, b_i$, we have by Eqs. (18,19) that when $E_{a,\varepsilon}$ or $E_{R,R/6}$ occur then the $i$-th neuron satisfies $|w_i| \geq \sigma/R$ and

$$|b_i| \leq (R + \frac{1}{6}R)w_i \leq \frac{7}{6}R \cdot \frac{2\sigma_h}{R} = \frac{7}{3}\sigma_h,$$

concluding the derivation of Items 1-4.

5. It will suffice to lower bound the probability of the event denoted by $A$ where $x_i \notin (\alpha_j - \gamma/2, \alpha_j + \gamma/2)$ for all $i \in [n]$ and $j \in [r]$, where $\alpha_j$ is the $j$-th sign change of the ground truth function labelling $y_i$. The set $\cup_{j=1}^{r}(\alpha_j - \gamma/2, \alpha_j + \gamma/2)$ has Lebesgue measure of at most $\gamma r$, and since by Assumption 2.2 we have that $\mu(x) \leq C$ for all $x \in [-R, R]$, we lower bound the probability of the event by the expression

$$\mathbb{P}[A] \geq (1 - \gamma rC)^n.$$

Plugging $\gamma = \frac{\delta}{4nrC}$ in the above which entails $\gamma rC \leq 1$ and using Bernoulli's inequality we have

$$\mathbb{P}[A] \geq 1 - \frac{\delta}{4}.$$

To conclude the derivation so far, using another union bound, we have shown that with probability at least $1 - 0.75\delta$, $\boldsymbol{\theta}(0)$ is separable from $S$ with constants

$$\gamma_0 = \frac{\delta}{4nrC}, \quad q_0 = \frac{\rho}{4}, \quad Q_0 = \frac{7}{3}R, \quad m_0 = \frac{\sigma_h}{R}, \quad M_0 = \frac{7}{3}\sigma_h. \tag{21}$$

We will now show that the separability also holds in a $\Delta$-hidden neighborhood of $\boldsymbol{\theta}(0)$ for an appropriately chosen $\Delta > 0$. To this end, we first establish that

$$\mathbb{P}\left[\min_{i \in [n], j \in [k]} |\beta_j - x_i| > \frac{\delta}{8nkC}\right] \geq 1 - \frac{\delta}{4}.$$

Suppose we have $n$ data instances in $[-R, R]$, then a cover of radius $\frac{\delta\pi}{8kn}$ over these data points has a (one-dimensional Lebesgue) measure of at most $\frac{\delta\pi}{4k}$. Thus, the probability that a breakpoint will not be initialized within distance less than $\frac{\delta\pi}{8kn}$ from any point is at least $(1 - \frac{\delta}{4k})$. This is true since Assumption 2.1 implies that the distribution of a breakpoint is a standard Cauchy distribution with density at most $1/\pi$. We thus have that

$$\mathbb{P}\left[\min_{i \in [n], j \in [k]} |\beta_j - x_i| > \frac{\delta}{8nkC}\right] \geq \left(1 - \frac{\delta}{4k}\right)^k \geq 1 - \frac{\delta}{4}, \tag{22}$$

where the last inequality follows from Bernoulli's inequality. A final union bound now implies that the above bound holds with the previous implications with probability at least $1 - \delta$. Define

$$\Delta := \frac{\delta\rho\sigma_h}{24nkCR^3},$$

we will now show that this implies the uniform separability of any $\boldsymbol{\theta} \in U_\Delta(\boldsymbol{\theta}(0))$ from $S$, by proving Items 1-4 jointly and Item 5 separately.

1-4. First, by Assumption 2.2, we have $1 = \int_{-R}^{R} \mu(x)dx \leq 2RC$ which with Eq. (18) implies that $\Delta \leq \frac{\delta\sigma_h}{24nkR} \leq \frac{\sigma_h}{24R}$. Since the weight and bias of each neuron in $U_\Delta(\boldsymbol{\theta}(0))$ change by at most $\Delta$, we have

$$m \geq \frac{\sigma_h}{R} - \frac{\sigma_h}{24R} = \frac{23\sigma_h}{24R} \quad \text{and} \quad M \leq \frac{7}{3}\sigma_h + \frac{\sigma_h}{24R} \leq \frac{57}{24}\sigma_h.$$

To bound $q$ and $Q$, we will first show that under our assumptions the breakpoints cannot move much. To this end, we show that for each neuron, the function $f(w, b) := -\frac{b}{w}$ is Lipschitz on $U_\Delta(\boldsymbol{\theta}(0))$. We have

$$\nabla f(w, b) = \left(\frac{b}{w^2}, -\frac{1}{w}\right),$$

and therefore for any neuron $(w, b) \in \boldsymbol{\theta}$ such that $\boldsymbol{\theta} \in U_\Delta(\boldsymbol{\theta}(0))$ we get

$$\|\nabla f(w, b)\| = \sqrt{\frac{b^2}{w^4} + \frac{1}{w^2}} \leq \sqrt{\frac{M^2}{m^4} + \frac{1}{m^2}} \leq \frac{24R}{23\sigma_{\mathrm{h}}}\sqrt{\frac{57^2}{23^2}R^2 + 1} < \frac{3R^2}{\sigma_{\mathrm{h}}},$$

where the last inequality follows from $1 \leq R^2$. This implies that

$$\left| -\frac{b}{w} + \frac{b_0}{w_0} \right| \leq \|\nabla f(w, b)\| \cdot \|(w, b) - (w_0, b_0)\| < \frac{3R^2}{\sigma_{\mathrm{h}}}\Delta \leq \frac{\delta\rho}{8nkCR}.$$

That is, we have that the breakpoint of each neuron moves a distance strictly less than $\frac{\delta\rho}{8nkCR} \leq \frac{\delta}{8nkC}$, which along with Eq. (22) guarantees that $\mathcal{L}(\cdot)$ is differentiable on $U_\Delta(\boldsymbol{\theta}(0))$ since no ReLU crosses a data instance. Since $Q$ is the upper bound on the difference between two breakpoints where each moves by at most $\frac{\delta}{8nkC}$, this also yields a bound on $Q$ as follows

$$Q \leq Q_0 + 2\frac{\delta}{8nkC} \leq \frac{7}{3}R + \frac{1}{8}R \leq 2.5R,$$

where we used the upper bound on $Q_0$ from Eq. (21), Eq. (17) which implies $k \geq 4$ (since $\rho \leq R$ by Eq. (18)), and $1/C \leq 2R$. Likewise, to lower bound $q$, compute

$$q \geq q_0 - 2\frac{\delta\rho}{8nkCR} \geq \frac{\rho}{4} - \frac{\rho}{20R} \geq \frac{\rho}{5},$$

where again we used Eq. (21), Eq. (17) which implies $k \geq 10$, and $1/C \leq 2R$.

5. Since $\gamma$ depends on $S$ and not on $\boldsymbol{\theta}$, it remains unchanged and we have $\gamma = \gamma_0$.

We can now use the assumption $\boldsymbol{\theta} \in L^+_{1/2n}(\mathcal{L})$ and Lemma C.2 to conclude

$$\frac{1}{2}\|\nabla\mathcal{L}(\boldsymbol{\theta})\|^2_2 \geq \frac{\gamma^2 q^2 m^6}{259200 n^4 Q^2 M^4} \geq \frac{1}{259200 n^4} \cdot \frac{\delta^2}{4^2 n^2 r^2 C^2} \cdot \frac{4\rho^2}{25^2 R^2} \cdot \frac{23^6}{57^4 24^2 R^6} \cdot \sigma^2_{\mathrm{h}}.$$

Simplifying the above, the proposition follows.

$\square$

Having established the required machinery for proving Theorem 3.1, we now turn to do so.

*Proof of Theorem 3.1.* We begin with bounding the loss upon initialization with high probability. First, consider $3k$ i.i.d. random variables $X_j \sim \mathcal{N}(0, 1)$. We have that

$$\mathbb{P}\left[\max_{j \in [3k]} |X_j| \leq x\right] = \left(\mathrm{erf}\left(\frac{x}{\sqrt{2}}\right)\right)^{3k} \geq \left(1 - \exp\left(-0.5x^2\right)\right)^{3k} \geq 1 - 3k\exp\left(-0.5x^2\right),$$

where the first inequality follows from $1 - \mathrm{erf}(x) < \exp(-x^2)$ for all $x \geq 0$ (see Eq. (7.8.3) in DLMF) and the second inequality follows from Bernoulli's inequality since $\exp(-0.5x^2) < 1$. Plugging $x = \sqrt{2\log(6k/\delta)}$ in the above, we have

$$\mathbb{P}\left[\max_{j \in [3k]} |X_j| \leq \sqrt{2\log(6k/\delta)}\right] \geq 1 - \frac{3k\delta}{6k} = 1 - \frac{\delta}{2}.$$

Thus, with probability at least $1 - \frac{\delta}{2}$, we have that all the weights of $\boldsymbol{\theta}(0)$ are at most $\sqrt{2\log(6k/\delta)}$ standard deviations away from zero. With this bound, we can derive for all $x \in [-R, R]$

$$\mathcal{N}_{\boldsymbol{\theta}(0)}(x) \leq \sum_{j \in [k]} |v_j|\sigma(|w_j| \cdot |x| + |b_j|) \leq 4kR\sigma_{\mathrm{h}}\sigma_{\mathrm{o}}\log\left(\frac{6k}{\delta}\right),$$

which for both the exponential and logistic losses implies

$$\mathcal{L}(\boldsymbol{\theta}(0)) \leq \exp\left(4kR\sigma_{\mathrm{h}}\sigma_{\mathrm{o}}\log\left(\frac{6k}{\delta}\right)\right) \leq e, \tag{23}$$

where the last inequality is by our assumption $\sigma_{\mathrm{o}} \leq \frac{1}{4kR\sigma_{\mathrm{h}}\log\left(\frac{6k}{\delta}\right)}$. Letting

$$\lambda := 10^{-11}\frac{\delta^2\rho^2}{n^6r^2C^2R^8}$$

and observing that our lower bound assumption on $\sigma_{\mathrm{h}}$ in Eq. (3) implies it's at least $0.5$ since $\rho \leq R$ and $C \geq 1/2R$, we can invoke Proposition C.1 with confidence $\frac{\delta}{2}$ to obtain

$$\begin{aligned}
\frac{1}{2}\left\|\nabla\mathcal{L}(\boldsymbol{\theta}(t))\right\|_2^2 &\geq 3\cdot 10^{-11}\frac{\delta^2\rho^2}{n^6r^2C^2R^8}\sigma_{\mathrm{h}}^2 \\
&\geq 3\cdot 10^{-11}\frac{\delta^2\rho^2}{n^6r^2C^2R^8}\sigma_{\mathrm{h}}^2\cdot\frac{\mathcal{L}(\boldsymbol{\theta}(t))}{\mathcal{L}(\boldsymbol{\theta}(0))} \\
&\geq \lambda\sigma_{\mathrm{h}}^2\cdot\mathcal{L}(\boldsymbol{\theta}(t)),
\end{aligned} \tag{24}$$

where the second inequality holds since $\frac{\mathcal{L}(\boldsymbol{\theta}(t))}{\mathcal{L}(\boldsymbol{\theta}(0))} \leq 1$ because the flow is non-increasing, and the last inequality holds due to Eq. (23) which implies $\frac{1}{\mathcal{L}(\boldsymbol{\theta}(0))} \geq \exp(-1) \geq \frac{1}{3}$. By a union bound, the above holds with probability at least $1 - \delta$.

Denote $D := U_\Delta(\boldsymbol{\theta}(0)) \cap L_{\frac{1}{2n}}^+(\mathcal{L})$ where $\Delta$ is defined in Eq. (17), and define $t' \in [0,\infty)$ to be the smallest time such that $\boldsymbol{\theta}(t')$ is on the boundary of $U_\Delta(\boldsymbol{\theta}(0))$ (where $t' = \infty$ if there exists no such time). We will now show that the flow attains loss at most $\frac{1}{2n}$ in time $t_0 := \frac{\log(2n\mathcal{L}(\boldsymbol{\theta}(0)))}{2\lambda\sigma_{\mathrm{h}}^2}$, by analyzing several different cases.

- Suppose that $t' > t_0$.
  - If $\{\boldsymbol{\theta}(t) : t \in [0,t_0]\} \subseteq D$, then by the PL-condition shown in Eq. (24) we have for all $t \in [0,t_0]$ that GF enjoys a convergence rate of

    $$\mathcal{L}(\boldsymbol{\theta}(t)) \leq \exp\left(-2\lambda\sigma_{\mathrm{h}}^2 t\right)\cdot\mathcal{L}(\boldsymbol{\theta}(0)).$$

    Plugging $t = t_0$ in the above and simplifying, we have $\mathcal{L}(\boldsymbol{\theta}(t_0)) \leq \frac{1}{2n}$.
  - If $\{\boldsymbol{\theta}(t) : t \in [0,t_0]\} \not\subseteq D$, then there exists a time $t'' \leq t_0$ such that $\boldsymbol{\theta}(t'') \notin D$. Since $t'' \leq t_0 < t'$, it must hold that $t'' \notin L_{\frac{1}{2n}}^+(\mathcal{L})$, and therefore $\mathcal{L}(\boldsymbol{\theta}(t'')) < \frac{1}{2n}$ which implies $\mathcal{L}(\boldsymbol{\theta}(t_0)) < \frac{1}{2n}$ since the flow is non-increasing.

- Suppose that $t' \leq t_0$. Assume by contradiction that $\{\boldsymbol{\theta}(t) : t \in [0,t']\} \subseteq D$. We will now show that the length of the trajectory of GF cannot have been long enough to reach the boundary of $D$, which will result in a contradiction. To this end, we use a similar technique as in Gupta et al. [2021, Thm. 9]. Define the potential function $\varepsilon(t) = \sqrt{\mathcal{L}(\boldsymbol{\theta}(t))}$. Taking the derivative of $\varepsilon(t)$ with respect to $t$ and using the chain rule we have

$$\dot{\varepsilon}(t) = \frac{\frac{d\mathcal{L}(\boldsymbol{\theta}(t))}{dt}}{2\sqrt{\mathcal{L}(\boldsymbol{\theta}(t))}} = -\frac{\left\|\nabla\mathcal{L}(\boldsymbol{\theta}(t))\right\|_2^2}{2\sqrt{\mathcal{L}(\boldsymbol{\theta}(t))}} \leq -\sqrt{\frac{\lambda}{2}}\sigma_{\mathrm{h}}\cdot\left\|\nabla\mathcal{L}(\boldsymbol{\theta}(t))\right\|_2,$$

where the inequality follows from Eq. (24). We can now bound the length of the trajectory up until time $t'$ by using the fundamental theorem of calculus and obtain

$$\begin{aligned}
\int_0^{t'}\left\|\nabla\mathcal{L}(\boldsymbol{\theta}(t))\right\|_2 dt &\leq -\frac{1}{\sigma_{\mathrm{h}}}\sqrt{\frac{2}{\lambda}}\int_0^{t'}\dot{\varepsilon}(t)dt \leq -\frac{1}{\sigma_{\mathrm{h}}}\sqrt{\frac{2}{\lambda}}\left[\sqrt{\mathcal{L}(\boldsymbol{\theta}(t))}\right]_0^{t'} \\
&\leq \frac{1}{\sigma_{\mathrm{h}}}\sqrt{\frac{2\mathcal{L}(\boldsymbol{\theta}(0))}{\lambda}} \leq \frac{1}{\sigma_{\mathrm{h}}}\sqrt{\frac{2e}{\lambda}} < \Delta,
\end{aligned}$$

where in the second line, the first inequality uses the fact that $\mathcal{L}(\cdot) > 0$, the second inequality follows from Eq. (23), and the last inequality follows from our bound on $\sigma_{\mathrm{h}}$ assumed in Eq. (3) and the definition of $\Delta$ in Eq. (17). In contrast, since $\boldsymbol{\theta}(t')$ is on the boundary of $U_\Delta(\boldsymbol{\theta}(0))$, this implies that there exists some neuron with weight and bias $w(t), b(t)$ at time

$t \geq 0$ such that $\|(w(t'), b(t')) - (w(0), b(0))\|_2 = \Delta$. From this and the path length upper bound we have

$$\Delta \leq \|\boldsymbol{\theta}(t') - \boldsymbol{\theta}(0)\|_2 \leq \int_0^{t'} \|\nabla \mathcal{L}(\boldsymbol{\theta}(t))\|_2 \, dt < \Delta,$$

which is a contradiction. We therefore must have that $\{\boldsymbol{\theta}(t) : t \in [0, t']\} \not\subseteq D$. When this holds, there exists a time $t'' \leq t' \leq t_0$ such that $\boldsymbol{\theta}(t'') \notin D$. Since $t'' \leq t'$, it must hold that $t'' \notin L^+_{\frac{1}{2n}}(\mathcal{L})$, and therefore $\mathcal{L}(\boldsymbol{\theta}(t'')) < \frac{1}{2n}$ which implies $\mathcal{L}(\boldsymbol{\theta}(t_0)) < \frac{1}{2n}$ since the flow is non-increasing.

$\square$

## D    Over-parameterization is necessary

In this appendix, we further discuss and formally prove Theorem 3.2, which establishes that in general under Assumption 2.1, an over-parameterization of magnitude at least $1.3r$ is necessary for achieving population loss below a constant. Our analysis is based on the following specific construction, where the labels are determined by a function $f_r$ parameterized by a natural number $r$ for all $x \in [-1, 1]$, expressible by the sign of a teacher network of width $r$ and defined as

$$f_r(x) \coloneqq \text{sign}(\sin(0.5\pi(r+1)(x+1))). \tag{25}$$

That is, $f_r$ changes value $r$ times between $-1$ and $1$ on the interval $[-1, 1]$, and is constant along intervals of length $\frac{2}{r+1}$. We now define the distribution $\mathcal{D}$ over the inputs of the dataset used in our lower bound and its corresponding labelling rule. We have

$$x \sim U[-1, 1] \quad \text{and} \quad y = f_r(x). \tag{26}$$

Recall the statement of Theorem 3.2, we have for example that if $\alpha = 1.3$, then the width of the network being trained is no more than $1.3r$ and GF attains loss at least $\frac{1}{160}$ in this case. While a lower bound of $r$ neurons for the construction specified in Eq. (26) is trivially implied by function approximation considerations, our lower bound merely improves upon this quantity by a constant multiplicative factor. Nevertheless, it is interesting to compare our lower bound to other similar settings in the literature, since it is typically difficult to derive lower bounds that require strictly more than $r$ neurons. For example, in a teacher-student setting where networks of the form $\mathbf{x} \mapsto \sum_{i=1}^r \sigma\left(\mathbf{w}_i^\top \mathbf{x}\right)$ are considered, it is known that there are spurious (non-global) minima already when $r \geq 6$ [Safran and Shamir, 2018, Arjevani and Field, 2020, 2021], and that empirically we are more likely to get stuck in those minima the larger $r$ is [Safran and Shamir, 2018], but in spite of this ample empirical evidence, there is no proof that optimization will fail for any natural number $r \geq 6$. In contrast, in our univariate setting, it is possible to show a non-trivial lower bound since we utilize bias terms. This highlights the difference between settings that omit and include biases, which impacts the associated optimization problem in a non-trivial manner.

The proof of our lower bound, which appears below in Appendix D.1, relies on the observation that under Assumption 2.1, the breakpoints of the trained network upon initialization follow a standard Cauchy distribution. In such a case, neurons with a breakpoint outside the support of the data and with the wrong orientation will remain dormant throughout the optimization process, which requires initializing at least a fraction more of the minimal number of neurons required so that sufficiently many will be optimized and could improve the approximation of the target function. While one can circumvent this issue by scaling the breakpoints to the support of the data, this would require (i) an initialization scheme which is different than Assumption 2.1, which is used in our upper bounds; and (ii) this may even prove detrimental to optimization, as our positive result requires neurons that are active on all the data instances. We stress that our lower bound given here applies to training over a sample of any size, since it relies on approximation arguments. Additionally, we remark that by scaling the distribution $\mathcal{D}$ to be supported on a smaller interval we can increase the required magnitude of over-parameterization up to a factor of $\alpha = 2$, however due to the common practice of scaling the data to have unit norm, we assume it is supported on $[-1, 1]$. We also remark that a common initialization scheme is to set the bias terms to zero [He et al., 2015]. This results in breakpoints that are initialized at the origin and circumvents the issue of dormant neurons upon

initialization, however the main motivation for using such an initialization scheme is to achieve numerical stability and avoid exploding gradients when training very deep networks, which is not an issue for the shallow architecture we consider here. In any case, we stress that the goal of our lower bounds is to exemplify that over-parameterization is necessary in a setting complementary to our upper bound in Theorem 3.1, and we leave the derivation of stronger lower bounds under more general initialization schemes as a tantalizing future work direction.

### D.1 Proof of Theorem 3.2

To prove the theorem, we would need the following auxiliary lemmas. The first lemma below establishes that if we approximate the function $f_r$ which is defined in Eq. (25) by a function which does not change its sign over an interval of length larger than $\frac{2}{r+1}$, then this results in a strictly positive loss which is roughly proportional to the length of the approximation interval.

**Lemma D.1.** *Let $f_r$ be as defined in Eq. (25), let $\beta_1, \beta_2 \in [-1, 1]$, $\mathcal{N} : [\beta_1, \beta_2] \to \mathbb{R}$ such that its sign is fixed on $[\beta_1, \beta_2]$, and let $\ell$ be either the exponential or logistic loss. Then*

$$\int_{\beta_1}^{\beta_2} \ell(\mathcal{N}(x) \cdot f_r(x)) dx \geq \frac{1}{2} \ell(0) \left( \beta_2 - \beta_1 - \frac{2}{r+1} \right).$$

*Proof.* If $\beta_2 - \beta_1 \leq \frac{2}{r+1}$, then the right-hand side is non-positive and the lemma follows since $\ell(\cdot) > 0$. If $\beta_2 - \beta_1 > \frac{2}{r+1}$, then the (one-dimensional Lebesgue) measure of the set $A := \{x \in [\beta_1, \beta_2] : \mathcal{N}(x) \cdot f_r(x) \leq 0\}$ is at least

$$\frac{1}{2} \left( \beta_2 - \beta_1 - \frac{2}{r+1} \right),$$

since the measure of the complementary set $\{x \in [\beta_1, \beta_2] : \mathcal{N}(x) \cdot f_r(x) > 0\}$ is at most $\frac{1}{2} \left( \beta_2 - \beta_1 + \frac{2}{r+1} \right)$, where the upper bound is attained when $\mathcal{N}(x) \cdot f_r(x) > 0$ for all

$$x \in \left[ \beta_1, \beta_1 + \frac{2}{r+1} \right] \cup \left[ \beta_2 - \frac{2}{r+1}, \beta_2 \right].$$

We can therefore lower bound the integral in the lemma by

$$\int_{\beta_1}^{\beta_2} \ell(\mathcal{N}(x) \cdot f_r(x)) dx \geq \int_{x \in A} \ell(\mathcal{N}(x) \cdot f_r(x)) dx \geq \int_{x \in A} \ell(0) dx \geq \frac{1}{2} \ell(0) \left( \beta_2 - \beta_1 - \frac{2}{r+1} \right).$$

$\square$

The following lemma shows that approximating $f_r$ using a ReLU network with just $r'$ neurons results in loss proportional to $1 - r'/r$.

**Lemma D.2.** *Suppose that $f_r$ as defined in Eq. (25). Then for any ReLU network $\mathcal{N}_{\boldsymbol{\theta}}$ of width at most $r'$, we have*

$$\mathcal{L}_{\mathcal{D}}(\boldsymbol{\theta}) \geq \frac{1}{4} \left( 1 - \frac{r'}{r} \right).$$

*Proof.* Denote by $\beta_1, \ldots, \beta_{r'}$ the set of points where $\mathcal{N}$ changes sign in $(-1, 1)$. Note that this set is always of size at most $r'$, and we may assume without loss of generality that it is of size exactly $r'$ (since otherwise we can prove a stronger claim, where the lemma holds for some $r'' < r'$). Further define the boundaries $\beta_0 := -1$ and $\beta_{r'+1} := 1$. We compute

$$\mathcal{L}_{\mathcal{D}}(\boldsymbol{\theta}) = \int_{-1}^{1} \frac{1}{2} \ell(f_r(x) \cdot \mathcal{N}_{\boldsymbol{\theta}}(x)) dx = \frac{1}{2} \sum_{i=0}^{r'} \int_{\beta_i}^{\beta_{i+1}} \ell(f_r(x) \cdot \mathcal{N}_{\boldsymbol{\theta}}(x)) dx$$

$$\geq \frac{1}{4} \ell(0) \sum_{i=0}^{r'} \left( \beta_{i+1} - \beta_i - \frac{2}{r+1} \right) = \frac{1}{4} \left( 2 - 2 \frac{r'+1}{r+1} \right)$$

$$= \frac{1}{2} \cdot \frac{r - r'}{r+1} \geq \frac{1}{4} \left( 1 - \frac{r'}{r} \right),$$

where the first inequality uses Lemma D.1, the equality that follows is due to the sum telescoping and since $\ell(0) = 1$ for both the exponential and logistic losses, and the final inequality is due to $r \geq 1$. □

With the above auxiliary lemmas, we can now turn to the proof of the theorem.

*Proof of Theorem 3.2.* By Assumption 2.1, the breakpoints of $\mathcal{N}$ at initialization follow a standard Cauchy distribution. Since $\mathcal{D}$ is supported on $[-1, 1]$, we have with probability exactly $0.5$ that the breakpoint of a given neuron falls outside of $[-1, 1]$. Moreover, with an independent probability of $0.5$, the orientation of the neuron is such that it is off on all the data instances. I.e., such a neuron remains dormant throughout the optimization process of GF with probability $0.25$. It can be verified that for any integer $k \geq 1$, at least $\lceil 0.25k \rceil$ neurons will be dormant upon initialization with probability at least $0.25$.[7] Thus, with probability at least $0.25$, we have that out of $\alpha r$ neurons, there are at most $r' = \lfloor 0.75\alpha r \rfloor$ neurons that are effectively being trained with breakpoints in $[-1, 1]$. By Lemma D.2, this results in a lower bound on the population loss of

$$\mathcal{L}_{\mathcal{D}}(\mathcal{N}_{\boldsymbol{\theta}(t)}) \geq \frac{1}{4}\left(1 - \frac{\lfloor 0.75\alpha r \rfloor}{r}\right) \geq \frac{1}{4}\left(1 - 0.75\alpha\right),$$

for any time $t \geq 0$. □

# E    Proof of Theorem 4.2

By Theorem 4.1, if there exists time $t_0$ such that $\mathcal{L}(\boldsymbol{\theta}(t_0)) < \frac{1}{n}$ then GF converges to zero loss, and converges in direction to a KKT point of Problem (4). We denote $\mathcal{N}_{\boldsymbol{\theta}}(x) = \sum_{j \in [k]} v_j \sigma(w_j x + b_j)$. Thus, $\mathcal{N}_{\boldsymbol{\theta}}$ is a network of width $k$, where the weights in the first layer are $w_1, \ldots, w_k$, the bias terms are $b_1, \ldots, b_k$, and the weights in the second layer are $v_1, \ldots, v_k$. We denote $J := \{j \in [k] : v_j \neq 0\}$, $J^+ := \{j \in J : v_j > 0\}$, and $J^- := \{j \in J : v_j < 0\}$. Since neurons with output weight $v_j = 0$ do not affect the function that the network computes, then in this proof we ignore them. We also denote $I := [n]$, and $I' = \{i \in I : y_i \mathcal{N}_{\boldsymbol{\theta}}(x_i) = 1\}$. Thus, $I'$ are the indices of the examples where $\mathcal{N}_{\boldsymbol{\theta}}$ attains margin of exactly $1$.

Assume that $\mathcal{N}_{\boldsymbol{\theta}}$ satisfies the KKT conditions of Problem (4). Thus, there are $\lambda_1, \ldots, \lambda_n$ such that for every $j \in J$ we have

$$w_j = \sum_{i \in I} \lambda_i \frac{\partial}{\partial w_j}\left(y_i \mathcal{N}_{\boldsymbol{\theta}}(x_i)\right) = \sum_{i \in I} \lambda_i y_i v_j \sigma'_{i,j} x_i \,, \tag{27}$$

where $\sigma'_{i,j}$ is a subgradient of $\sigma$ at $w_j \cdot x_i + b_j$, i.e., if $w_j \cdot x_i + b_j \neq 0$ then $\sigma'_{i,j} = \mathbb{1}[w_j \cdot x_i + b_j > 0]$, and otherwise $\sigma'_{i,j}$ is some value in $[0, 1]$ (we emphasize that in this case $\sigma'_{i,j}$ may be any value in $[0, 1]$ and in this proof we do not have any further assumptions on it). Also we have $\lambda_i \geq 0$ for all $i \in I$, and $\lambda_i = 0$ if $i \notin I'$. Likewise, we have

$$b_j = \sum_{i \in I} \lambda_i \frac{\partial}{\partial b_j}\left(y_i \mathcal{N}_{\boldsymbol{\theta}}(x_i)\right) = \sum_{i \in I} \lambda_i y_i v_j \sigma'_{i,j} \,. \tag{28}$$

We say that $\mathcal{N}_{\boldsymbol{\theta}}$ has an *activation point* at $x$ if there is $j \in [J]$ with $w_j \neq 0$ such that $w_j \cdot x + b_j = 0$. In this case we say that *the activation point $x$ corresponds to the neuron $j$*. Note that if $w_j = 0$ then the neuron computes a constant function and thus it does not affect the number of linear regions in $\mathcal{N}_{\boldsymbol{\theta}}$.

**Lemma E.1.** *We denote $I' = \{i_1, \ldots, i_q\}$ where $1 \leq i_1 < \ldots < i_q \leq [n]$. For every $\ell \in [q-1]$ the network $\mathcal{N}_{\boldsymbol{\theta}}$ has at most two activation points in the open interval $(x_{i_\ell}, x_{i_{\ell+1}})$. Moreover, $\mathcal{N}_{\boldsymbol{\theta}}$ has at most one activation point in $(-\infty, i_1)$ and at most one activation points in $(i_q, \infty)$.*

---

[7] Essentially, this holds true since the median of a binomially-distributed random variable $B(n, k)$ is $\lceil nk \rceil$ or $\lfloor nk \rfloor$, and since deviating from the median by at most 1 never increases the probability of the tail to more than $0.75$.

*Proof.* Let $x \in (x_{i_\ell}, x_{i_{\ell+1}})$ be an activation point, and let $j \in J$ such that $w_j \neq 0$ and $w_j \cdot x + b_j = 0$.

Suppose first that $w_j > 0$. Since $w_j \cdot x + b_j = 0$ then for every $x' > x$ we have $w_j \cdot x' + b_j > 0$, and for every $x' < x$ we have $w_j \cdot x' + b_j < 0$. By Eq. (27) we have

$$w_j = \sum_{i \in I} \lambda_i y_i v_j \sigma'_{i,j} x_i = \sum_{i \in I'} \lambda_i y_i v_j \sigma'_{i,j} x_i \, .$$

Since $x \neq x_i$ for all $i \in I'$ then $w_j \cdot x_i + b_j \neq 0$, and we have $\sigma'_{i,j} = \mathbb{1}(w_j \cdot x_i + b_j > 0) = \mathbb{1}[x_i > x]$. Therefore, the above displayed equation equals

$$\sum_{i \in I'} \lambda_i y_i v_j \mathbb{1}[x_i > x] x_i = \sum_{i \in I', i \geq i_{\ell+1}} \lambda_i y_i v_j x_i \, . \tag{29}$$

Likewise, by Eq. (28) we have

$$b_j = \sum_{i \in I} \lambda_i y_i v_j \sigma'_{i,j} = \sum_{i \in I'} \lambda_i y_i v_j \sigma'_{i,j} = \sum_{i \in I'} \lambda_i y_i v_j \mathbb{1}[x_i > x] = \sum_{i \in I', i \geq i_{\ell+1}} \lambda_i y_i v_j \, . \tag{30}$$

By Eq. (29) and Eq. (30), the activation point $x$ satisfies

$$x = \frac{-b_j}{w_j} = \frac{-\sum_{i \in I', i \geq i_{\ell+1}} \lambda_i y_i v_j}{\sum_{i \in I', i \geq i_{\ell+1}} \lambda_i y_i v_j x_i} = \frac{-\sum_{i \in I', i \geq i_{\ell+1}} \lambda_i y_i}{\sum_{i \in I', i \geq i_{\ell+1}} \lambda_i y_i x_i} \, .$$

Therefore if $x$ and $x'$ are two activation points in $(x_{i_\ell}, x_{i_{\ell+1}})$ that correspond to $w_j > 0$ and $w_{j'} > 0$ respectively, then $x = x'$. Thus, there is at most one activation point $x \in (x_{i_\ell}, x_{i_{\ell+1}})$ that corresponds to some $w_j > 0$.

Moreover, by similar arguments there is at most one activation point $x \in (x_{i_\ell}, x_{i_{\ell+1}})$ that corresponds to $w_j < 0$. Overall, in the interval $(x_{i_\ell}, x_{i_{\ell+1}})$ there are at most two activation points.

If $x \in (-\infty, i_1)$ then from similar argument we get that there is at most one activation point that corresponds to a neuron $j$ with $w_j > 0$. Also, an activation point in $(-\infty, i_1)$ that corresponds to a neuron with $w_j < 0$ does not exist, since such neuron is not active for any input $x_i$ with $i \in I'$, and hence by Eq. (27) we must have $w_j = 0$. The proof of the claim for the interval $(i_q, \infty)$ is similar. $\qquad\square$

Let $1 \leq a < b \leq n$ be indices such that for every $a \leq i < i' \leq b$ we have $y_i = y_{i'}$. Thus the labels do not switch signs for the inputs $x_a, x_{a+1}, \ldots, x_b$. Intuitively, the proof follows by showing that in the interval $[x_a, x_b]$ the network $\mathcal{N}_\theta$ has a constant number of linear regions, and then concluding that the overall number of linear regions in $\mathcal{N}_\theta$ must be $\mathcal{O}(r)$. We first consider the case where $x_b > x_a \geq 0$ and for all $a \leq i \leq b$ we have $y_i = 1$. In the following lemmas we analyze the activation points in this case and obtain a bound on the number of linear regions. Then, we will extend this result also to the cases where $y_i = -1$ and where $x_a < x_b \leq 0$. For a given activation point $x$ we say that the derivative of the network increases (respectively, decreases) in $x$ if for every sufficiently small $\varepsilon > 0$ the derivative of the network at $x - \varepsilon$ is smaller (respectively, larger) than the derivative at $x + \varepsilon$.

**Lemma E.2.** *Suppose that $x_b > x_a \geq 0$ and for all $a \leq i \leq b$ we have $y_i = 1$. In the interval $[x_a, x_b]$ the network $\mathcal{N}_\theta$ has at most two activation points where the derivative decreases.*

*Proof.* If the derivative decreases at an activation point $x \geq 0$, then there is at least one neuron $j \in J$ where $w_j \cdot x + b_j = 0$ and the derivative (of the function computed by this neuron) decreases in $x$. There are two types of such neurons: either (a) $w_j > 0$, $b_j \leq 0$ and $v_j < 0$; or (b) $w_j < 0$, $b_j \geq 0$ and $v_j < 0$.

We now show that in the interval $[x_a, x_b]$ there is at most one activation point that corresponds to a neuron of type (a) and at most one activation point that corresponds to a neuron of type (b). We note that an activation point might correspond to multiple neurons, namely, to a set $Q \subseteq J$ of neurons of size larger than 1. However, we show that if $x, x'$ are activation points in $[x_a, x_b]$ that correspond to sets $Q_x$ and $Q_{x'}$ of neurons (respectively) and both sets $Q_x, Q_{x'}$ contain neurons of type (a) then $x = x'$. Likewise, if both sets $Q_x, Q_{x'}$ contain neurons of type (b) then we also have $x = x'$.

Suppose towards contradiction that $x \in [x_a, x_b]$ is an activation point that corresponds to a neuron $j$ of type (a), and $x' \in [x_a, x_b]$ is an activation point with $x' > x$ that corresponds to a neuron $j'$

of type (a). Since both neurons $j, j'$ are of type (a), then we have $w_j \cdot z + b_j > 0$ iff $z > x$ and $w_{j'} \cdot z + b_{j'} > 0$ iff $z > x'$. By Eq. (27) we have

$$\frac{1}{v_{j'}} \cdot w_{j'} = \frac{1}{v_{j'}} \left( \sum_{i \in I} \lambda_i y_i v_{j'} \sigma'_{i,j'} x_i \right) = \sum_{i \in I} \lambda_i y_i \sigma'_{i,j'} x_i \le \sum_{i \in I} \lambda_i y_i \mathbb{1}[x_i \ge x'] x_i \,,$$

where the last inequality is since $\sigma'_{i,j'} = \mathbb{1}[x_i \ge x']$ if $x_i \ne x'$, and when $x_i = x'$ we have $\sigma'_{i,j'} \le \mathbb{1}[x_i \ge x']$ (and recall that for $a \le i \le b$ we have $y_i = 1$, $x_i \ge 0$ and $\lambda_i \ge 0$). The above RHS equals

$$\sum_{i \in I} \lambda_i y_i \mathbb{1}[x_i > x] x_i - \sum_{i \in I} \lambda_i y_i \mathbb{1}[x < x_i < x'] x_i \le \sum_{i \in I} \lambda_i y_i \mathbb{1}[x_i > x] x_i$$

$$\le \sum_{i \in I} \lambda_i y_i \sigma'_{i,j} x_i$$

$$= \frac{1}{v_j} \left( \sum_{i \in I} \lambda_i y_i v_j \sigma'_{i,j} x_i \right)$$

$$= \frac{1}{v_j} \cdot w_j \,.$$

Since $v_j < 0$ we conclude that

$$w_j \le \frac{v_j}{v_{j'}} \cdot w_{j'} \,. \tag{31}$$

Likewise, by Eq. (28) we have

$$\frac{1}{v_{j'}} \cdot b_{j'} = \frac{1}{v_{j'}} \left( \sum_{i \in I} \lambda_i y_i v_{j'} \sigma'_{i,j'} \right) = \sum_{i \in I} \lambda_i y_i \sigma'_{i,j'} \le \sum_{i \in I} \lambda_i y_i \mathbb{1}[x_i \ge x']$$

$$= \sum_{i \in I} \lambda_i y_i \mathbb{1}[x_i > x] - \sum_{i \in I} \lambda_i y_i \mathbb{1}[x < x_i < x']$$

$$\le \sum_{i \in I} \lambda_i y_i \sigma'_{i,j} = \frac{1}{v_j} \left( \sum_{i \in I} \lambda_i y_i v_j \sigma'_{i,j} \right) = \frac{1}{v_j} \cdot b_j \,.$$

Hence, we conclude that

$$b_j \le \frac{v_j}{v_{j'}} \cdot b_{j'} \,. \tag{32}$$

Since $0 \le x < x'$, then by using Eq. (31) and (32) we have

$$0 < w_j \cdot x' + b_j \le \frac{v_j}{v_{j'}} \cdot w_{j'} \cdot x' + \frac{v_j}{v_{j'}} \cdot b_{j'} = \frac{v_j}{v_{j'}} \left( w_{j'} \cdot x' + b_{j'} \right) = 0 \,.$$

Thus, we reached a contradiction.

Next, suppose that $x \in [x_a, x_b]$ is an activation point that corresponds to a neuron $j$ of type (b), and $x' \in [x_a, x_b]$ is an activation point with $x' > x$ that corresponds to a neuron $j'$ of type (b). We will reach a contradiction using similar arguments to the case of type (a) neurons, with some required modifications.

Since both neurons $j, j'$ are of type (b), then we have $w_j \cdot z + b_j > 0$ iff $z < x$ and $w_{j'} \cdot z + b_{j'} > 0$ iff $z < x'$. By Eq. (27) we have

$$\frac{1}{v_{j'}} \cdot w_{j'} = \frac{1}{v_{j'}} \left( \sum_{i \in I} \lambda_i y_i v_{j'} \sigma'_{i,j'} x_i \right) = \sum_{i \in I} \lambda_i y_i \sigma'_{i,j'} x_i \ge \sum_{i \in I} \lambda_i y_i \mathbb{1}[x_i < x'] x_i \,,$$

where the last inequality is since $\sigma'_{i,j'} = \mathbb{1}[x_i < x']$ if $x_i \neq x'$, and when $x_i = x'$ we have $\sigma'_{i,j'} \geq \mathbb{1}[x_i < x']$. The above RHS equals

$$\sum_{i \in I} \lambda_i y_i \mathbb{1}[x_i \leq x]x_i + \sum_{i \in I} \lambda_i y_i \mathbb{1}[x < x_i < x']x_i \geq \sum_{i \in I} \lambda_i y_i \mathbb{1}[x_i \leq x]x_i$$

$$\geq \sum_{i \in I} \lambda_i y_i \sigma'_{i,j}x_i$$

$$= \frac{1}{v_j}\left(\sum_{i \in I} \lambda_i y_i v_j \sigma'_{i,j}x_i\right)$$

$$= \frac{1}{v_j} \cdot w_j .$$

Since $v_j < 0$ we conclude that

$$w_j \geq \frac{v_j}{v_{j'}} \cdot w_{j'} . \tag{33}$$

Likewise, by Eq. (28) we have

$$\frac{1}{v_{j'}} \cdot b_{j'} = \frac{1}{v_{j'}}\left(\sum_{i \in I} \lambda_i y_i v_{j'} \sigma'_{i,j'}\right) = \sum_{i \in I} \lambda_i y_i \sigma'_{i,j'} \geq \sum_{i \in I} \lambda_i y_i \mathbb{1}[x_i < x']$$

$$= \sum_{i \in I} \lambda_i y_i \mathbb{1}[x_i \leq x] + \sum_{i \in I} \lambda_i y_i \mathbb{1}[x < x_i < x']$$

$$\geq \sum_{i \in I} \lambda_i y_i \sigma'_{i,j} = \frac{1}{v_j}\left(\sum_{i \in I} \lambda_i y_i v_j \sigma'_{i,j}\right) = \frac{1}{v_j} \cdot b_j .$$

Hence, we conclude that

$$b_j \geq \frac{v_j}{v_{j'}} \cdot b_{j'} . \tag{34}$$

Since $0 \leq x < x'$, and by using Eq. (33) and (34), we have

$$0 > w_j \cdot x' + b_j \geq \frac{v_j}{v_{j'}} \cdot w_{j'} \cdot x' + \frac{v_j}{v_{j'}} \cdot b_{j'} = \frac{v_j}{v_{j'}}\left(w_{j'} \cdot x' + b_{j'}\right) = 0 .$$

Thus, we reached a contradiction. $\qquad \square$

We denote $\mathcal{I}_{a,b} := \{a, a+1, \ldots, b\} \subseteq I$ and $\mathcal{I}'_{a,b} := \{i \in \mathcal{I}_{a,b} : y_i \mathcal{N}_{\theta}(x_i) = 1\} = \mathcal{I}_{a,b} \cap I'$. Thus, $\mathcal{I}'_{a,b}$ are the indices of the examples in the interval $[x_a, x_b]$ where $\mathcal{N}_{\theta}$ attains margin of exactly 1. We denote $\mathcal{I}'_{a,b} = \{i_1, \ldots, i_m\}$, where $a \leq i_1 < \ldots < i_m \leq b$.

**Lemma E.3.** *Suppose that $x_b > x_a \geq 0$ and for all $a \leq i \leq b$ we have $y_i = 1$. There are at most 2 indices $\ell \in [m-1]$ such that $\mathcal{N}_{\theta}(x) > 1$ for some $x \in [x_{i_\ell}, x_{i_{\ell+1}}]$.*

*Proof.* Assume that $\mathcal{N}_{\theta}(x) > 1$ for $x \in [x_{i_\ell}, x_{i_{\ell+1}}]$. Since $\mathcal{N}_{\theta}(x_{i_\ell}) = \mathcal{N}_{\theta}(x_{i_{\ell+1}}) = 1$, then we have $x \in (x_{i_\ell}, x_{i_{\ell+1}})$. Now, since $\mathcal{N}_{\theta}(x_{i_\ell}) = \mathcal{N}_{\theta}(x_{i_{\ell+1}}) = 1$ and $\mathcal{N}_{\theta}(x) > 1$ for some $x \in (x_{i_\ell}, x_{i_{\ell+1}})$, then there must be an activation point in $(x_{i_\ell}, x_{i_{\ell+1}})$ where the derivative decreases. Since by Lemma E.2 there are at most two such activation points in $[x_a, x_b]$ then the lemma follows. $\qquad \square$

**Lemma E.4.** *Suppose that $x_b > x_a \geq 0$ and for all $a \leq i \leq b$ we have $y_i = 1$. There are at most 5 indices $\ell \in [m-1]$ such that $\mathcal{N}_{\theta}(x) < 1$ for some $x \in [x_{i_\ell}, x_{i_{\ell+1}}]$.*

*Proof.* Assume that $\mathcal{N}_{\theta}(x) < 1$ for $x \in [x_{i_\ell}, x_{i_{\ell+1}}]$. Since $\mathcal{N}_{\theta}(x_{i_\ell}) = \mathcal{N}_{\theta}(x_{i_{\ell+1}}) = 1$, then $x \in (x_{i_\ell}, x_{i_{\ell+1}})$. If $\ell \neq m-1$ then we have $\mathcal{N}_{\theta}(x) < 1$ and $\mathcal{N}_{\theta}(x_{i_{\ell+1}}) = \mathcal{N}_{\theta}(x_{i_{\ell+2}}) = 1$. Hence, the interval $(x, x_{i_{\ell+1}})$ contains a point with positive derivative, and the interval $(x_{i_{\ell+1}}, x_{i_{\ell+2}})$ contains a point with non-positive derivative. Thus, there must be an activation point in $(x, x_{i_{\ell+2}})$ where the derivative decreases. Therefore, there is an activation point with decreasing derivative either in the interval $(x_{i_\ell}, x_{i_{\ell+1}}]$ or in the interval $[x_{i_{\ell+1}}, x_{i_{\ell+2}})$ (and possibly in both). Thus, an interval

$[x_{i_\ell}, x_{i_{\ell+1}}]$ for $\ell \neq m - 1$ might contain some $x$ with $\mathcal{N}_\theta(x) < 1$ only if there is an activation point with decreasing derivative in $(x_{i_\ell}, x_{i_{\ell+1}}]$ or $[x_{i_{\ell+1}}, x_{i_{\ell+2}})$. Since by Lemma E.2 there are at most two such activation points in $[x_a, x_b]$, then there are at most 4 intervals $[x_{i_\ell}, x_{i_{\ell+1}}]$ with $\ell \neq m - 1$ that contain some $x$ with $\mathcal{N}_\theta(x) < 1$. The interval $[x_{i_{m-1}}, x_{i_m}]$ might also contain such $x$. Overall, there are at most 5 indices $\ell \in [m-1]$ such that $\mathcal{N}_\theta(x) < 1$ for some $x \in [x_{i_\ell}, x_{i_{\ell+1}}]$. $\qquad\square$

**Lemma E.5.** *Suppose that $x_b > x_a \geq 0$ and for all $a \leq i \leq b$ we have $y_i = 1$. There are at most 30 boundaries between linear regions in $[x_{i_1}, x_{i_m}]$.*

*Proof.* By Lemmas E.3 and E.4 there are at most 7 indices $\ell \in [m-1]$ such that $\mathcal{N}_\theta(x) \neq 1$ for some $x \in [x_{i_\ell}, x_{i_{\ell+1}}]$. We denote the set of these indices by $R$. Let $x \in (x_{i_1}, x_{i_m})$ be a boundary between two linear regions. Note that if $x \in (x_{i_\ell}, x_{i_{\ell+1}})$ for some $\ell \in [m-1]$ then $\ell \in R$. Also, if $x = x_{i_\ell}$ for some $2 \leq \ell \leq m - 1$ then either $\ell \in R$ or $\ell - 1 \in R$. In any case, we have $x \in [x_{i_\ell}, x_{i_{\ell+1}}]$ for some $\ell \in R$. Note that each boundary between linear regions is also an activation point. Therefore, the number of boundaries between linear regions in $(x_{i_1}, x_{i_m})$ is at most the number of activation points in the intervals $[x_{i_\ell}, x_{i_{\ell+1}}]$ with $\ell \in R$. By Lemma E.1 each interval $[x_{i_\ell}, x_{i_{\ell+1}}]$ contains at most 4 activation points: two points in $(x_{i_\ell}, x_{i_{\ell+1}})$ and two in $\{x_{i_\ell}, x_{i_{\ell+1}}\}$. Overall, there are at most $|R| \cdot 4 \leq 28$ boundaries between linear regions in $(x_{i_1}, x_{i_m})$. Thus, there are at most 30 boundaries between linear regions in $[x_{i_1}, x_{i_m}]$. $\qquad\square$

In the above lemmas we considered the case where $x_b > x_a \geq 0$ and for all $a \leq i \leq b$ we have $y_i = 1$, and proved that $\mathcal{I}'_{a,b}$ is such that there are at most 30 boundaries between linear regions in $[x_{i_1}, x_{i_m}]$. In Subsection E.1 we show analogous results for the case where $x_b > x_a \geq 0$ and for all $a \leq i \leq b$ we have $y_i = -1$. Thus, if $x_b > x_a \geq 0$ and the labels do not switch sign in the interval $[x_a, x_b]$ (i.e., either all labels are 1 or all labels are $-1$) then there are at most 30 boundaries between linear regions in $[x_{i_1}, x_{i_m}]$. The case where $x_a < x_b \leq 0$ (and the labels do not switch sign in the interval $[x_a, x_b]$) can be handled in a similar manner. Thus, even where the inputs are negative, $\mathcal{I}'_{a,b}$ is such that there are at most 30 boundaries between linear regions in $[x_{i_1}, x_{i_m}]$. The proof for this case is similar and for conciseness we do not repeat it.

We are now ready to finish the proof of the theorem. Consider the set $I'$ of indices where $\mathcal{N}_\theta$ attains margin 1 and denote $I' = \{i_1, \ldots, i_q\}$. Note that if $I'$ is an empty set, then by Eq. (27) and (28) all neurons have $w_j = b_j = 0$ and hence the network $\mathcal{N}_\theta$ is the zero function. Let $\ell \leq \ell'$ be such that the labels of the examples in the dataset do not change sign in the interval $[x_{i_\ell}, x_{i_{\ell'}}]$, and either $0 \leq x_{i_\ell} \leq x_{i_{\ell'}}$ or $x_{i_\ell} \leq x_{i_{\ell'}} \leq 0$. Thus, the interval $[x_{i_\ell}, x_{i_{\ell'}}]$ contains at most 30 boundaries between linear regions. Also, by Lemma E.1 the interval $(x_{i_{\ell-1}}, x_{i_\ell})$ (or $(-\infty, x_{i_\ell})$ if $\ell = 1$) contains at most two boundaries between linear regions. Likewise, the interval $(x_{i_{\ell'}}, x_{i_{\ell'+1}})$ (or $(x_{i_{\ell'}}, \infty)$ if $\ell' = q$) contains at most two boundaries between linear regions. Recall that the labels in the dataset switch sign at most $r$ times. Overall, we get that the number of boundaries between linear regions in the whole domain $\mathbb{R}$ is at most $30(r+2) + 2(r+3) = 32r + 66$. Indeed, if one of the $r + 1$ intervals where $\mathcal{N}_\theta$ do not switch sign contains 0 then we split it into two intervals, and thus we obtain $r + 2$ intervals. Each of these intervals includes at most 30 boundaries, and outside of these intervals there are at most $2(r+3)$ boundaries. Thus, that are at most $32r + 67$ linear regions.

## E.1  Lemmas for the case $y_i = -1$

**Lemma E.6.** *Suppose that $x_b > x_a \geq 0$ and for all $a \leq i \leq b$ we have $y_i = -1$. In the interval $[x_a, x_b]$ the network $\mathcal{N}_\theta$ has at most two activation points where the derivative increases.*

*Proof.* If the derivative increases at an activation point $x \geq 0$, then there is a least one neuron $j \in J$ where $w_j \cdot x + b_j = 0$ and the derivative (of the function computed by this neuron) increases in $x$. There are two types of such neurons: either (a) $w_j > 0$, $b_j \leq 0$ and $v_j > 0$; or (b) $w_j < 0$, $b_j \geq 0$ and $v_j > 0$.

We now show that in the interval $[x_a, x_b]$ there is at most one activation point that corresponds to a neuron of type (a) and at most one activation point that corresponds to a neuron of type (b). We note that an activation point might correspond to multiple neurons, namely, to a set $Q \subseteq J$ of neurons of size larger than 1. However, we show that if $x, x'$ are activation points in $[x_a, x_b]$ that correspond to sets $Q_x$ and $Q_{x'}$ of neurons (respectively) and both sets $Q_x, Q_{x'}$ contain neurons of type (a) then $x = x'$. Likewise, if both sets $Q_x, Q_{x'}$ contain neurons of type (b) then we also have $x = x'$.

Suppose towards contradiction that $x \in [x_a, x_b]$ is an activation point that corresponds to a neuron $j$ of type (a), and $x' \in [x_a, x_b]$ is an activation point with $x' > x$ that corresponds to a neuron $j'$ of type (a). Since both neurons $j, j'$ are of type (a), then we have $w_j \cdot z + b_j > 0$ iff $z > x$ and $w_{j'} \cdot z + b_{j'} > 0$ iff $z > x'$. By Eq. (27) we have

$$\frac{1}{v_{j'}} \cdot w_{j'} = \frac{1}{v_{j'}} \left( \sum_{i \in I} \lambda_i y_i v_{j'} \sigma'_{i,j'} x_i \right) = \sum_{i \in I} \lambda_i y_i \sigma'_{i,j'} x_i \geq \sum_{i \in I} \lambda_i y_i \mathbb{1}[x_i \geq x'] x_i ,$$

where the last inequality is since $\sigma'_{i,j'} = \mathbb{1}[x_i \geq x']$ if $x_i \neq x'$, and when $x_i = x'$ we have $\sigma'_{i,j'} \leq \mathbb{1}[x_i \geq x']$ (and recall that for $a \leq i \leq b$ we have $y_i = -1$, $x_i \geq 0$ and $\lambda_i \geq 0$). The above RHS equals

$$\sum_{i \in I} \lambda_i y_i \mathbb{1}[x_i > x] x_i - \sum_{i \in I} \lambda_i y_i \mathbb{1}[x < x_i < x'] x_i \geq \sum_{i \in I} \lambda_i y_i \mathbb{1}[x_i > x] x_i$$

$$\geq \sum_{i \in I} \lambda_i y_i \sigma'_{i,j} x_i$$

$$= \frac{1}{v_j} \left( \sum_{i \in I} \lambda_i y_i v_j \sigma'_{i,j} x_i \right)$$

$$= \frac{1}{v_j} \cdot w_j .$$

We conclude that

$$w_j \leq \frac{v_j}{v_{j'}} \cdot w_{j'} . \tag{35}$$

Likewise, by Eq. (28) we have

$$\frac{1}{v_{j'}} \cdot b_{j'} = \frac{1}{v_{j'}} \left( \sum_{i \in I} \lambda_i y_i v_{j'} \sigma'_{i,j'} \right) = \sum_{i \in I} \lambda_i y_i \sigma'_{i,j'} \geq \sum_{i \in I} \lambda_i y_i \mathbb{1}[x_i \geq x']$$

$$= \sum_{i \in I} \lambda_i y_i \mathbb{1}[x_i > x] - \sum_{i \in I} \lambda_i y_i \mathbb{1}[x < x_i < x']$$

$$\geq \sum_{i \in I} \lambda_i y_i \sigma'_{i,j} = \frac{1}{v_j} \left( \sum_{i \in I} \lambda_i y_i v_j \sigma'_{i,j} \right) = \frac{1}{v_j} \cdot b_j .$$

Hence, we conclude that

$$b_j \leq \frac{v_j}{v_{j'}} \cdot b_{j'} . \tag{36}$$

Since $0 \leq x < x'$, then by using Eq. (35) and (36) we have

$$0 < w_j \cdot x' + b_j \leq \frac{v_j}{v_{j'}} \cdot w_{j'} \cdot x' + \frac{v_j}{v_{j'}} \cdot b_{j'} = \frac{v_j}{v_{j'}} (w_{j'} \cdot x' + b_{j'}) = 0 .$$

Thus, we reached a contradiction.

Next, suppose that $x \in [x_a, x_b]$ is an activation point that corresponds to a neuron $j$ of type (b), and $x' \in [x_a, x_b]$ is an activation point with $x' > x$ that corresponds to a neuron $j'$ of type (b). We will reach a contradiction using similar arguments to the case of type (a) neurons, with some required modifications.

Since both neurons $j, j'$ are of type (b), then we have $w_j \cdot z + b_j > 0$ iff $z < x$ and $w_{j'} \cdot z + b_{j'} > 0$ iff $z < x'$. By Eq. (27) we have

$$\frac{1}{v_{j'}} \cdot w_{j'} = \frac{1}{v_{j'}} \left( \sum_{i \in I} \lambda_i y_i v_{j'} \sigma'_{i,j'} x_i \right) = \sum_{i \in I} \lambda_i y_i \sigma'_{i,j'} x_i \leq \sum_{i \in I} \lambda_i y_i \mathbb{1}[x_i < x'] x_i ,$$

where the last inequality is since $\sigma'_{i,j'} = \mathbb{1}[x_i < x']$ if $x_i \neq x'$, and when $x_i = x'$ we have $\sigma'_{i,j'} \geq \mathbb{1}[x_i < x']$ (and $y_i = -1$). The above RHS equals

$$\sum_{i \in I} \lambda_i y_i \mathbb{1}[x_i \leq x]x_i + \sum_{i \in I} \lambda_i y_i \mathbb{1}[x < x_i < x']x_i \leq \sum_{i \in I} \lambda_i y_i \mathbb{1}[x_i \leq x]x_i$$

$$\leq \sum_{i \in I} \lambda_i y_i \sigma'_{i,j} x_i$$

$$= \frac{1}{v_j} \left( \sum_{i \in I} \lambda_i y_i v_j \sigma'_{i,j} x_i \right)$$

$$= \frac{1}{v_j} \cdot w_j .$$

We conclude that

$$w_j \geq \frac{v_j}{v_{j'}} \cdot w_{j'} . \tag{37}$$

Likewise, by Eq. (28) we have

$$\frac{1}{v_{j'}} \cdot b_{j'} = \frac{1}{v_{j'}} \left( \sum_{i \in I} \lambda_i y_i v_{j'} \sigma'_{i,j'} \right) = \sum_{i \in I} \lambda_i y_i \sigma'_{i,j'} \leq \sum_{i \in I} \lambda_i y_i \mathbb{1}[x_i < x']$$

$$= \sum_{i \in I} \lambda_i y_i \mathbb{1}[x_i \leq x] + \sum_{i \in I} \lambda_i y_i \mathbb{1}[x < x_i < x']$$

$$\leq \sum_{i \in I} \lambda_i y_i \sigma'_{i,j} = \frac{1}{v_j} \left( \sum_{i \in I} \lambda_i y_i v_j \sigma'_{i,j} \right) = \frac{1}{v_j} \cdot b_j .$$

Hence, we conclude that

$$b_j \geq \frac{v_j}{v_{j'}} \cdot b_{j'} . \tag{38}$$

Since $0 \leq x < x'$, and by using Eq. (37) and (38), we have

$$0 > w_j \cdot x' + b_j \geq \frac{v_j}{v_{j'}} \cdot w_{j'} \cdot x' + \frac{v_j}{v_{j'}} \cdot b_{j'} = \frac{v_j}{v_{j'}} \left( w_{j'} \cdot x' + b_{j'} \right) = 0 .$$

Thus, we reached a contradiction. $\qquad\square$

We use the notations $\mathcal{I}_{a,b} := \{a, a+1, \ldots, b\} \subseteq I$ and $\mathcal{I}'_{a,b} := \{i \in \mathcal{I}_{a,b} : y_i \mathcal{N}_{\boldsymbol{\theta}}(x_i) = 1\} = \mathcal{I} \cap I'$. Thus, $\mathcal{I}'$ are the indices of the examples in the interval $[x_a, x_b]$ where $\mathcal{N}_{\boldsymbol{\theta}}$ attains margin of exactly 1. We denote $\mathcal{I}'_{a,b} = \{i_1, \ldots, i_m\}$, where $a \leq i_1 < \ldots < i_m \leq b$.

**Lemma E.7.** *Suppose that $x_b > x_a \geq 0$ and for all $a \leq i \leq b$ we have $y_i = -1$. There are at most 2 indices $\ell \in [m-1]$ such that $\mathcal{N}_{\boldsymbol{\theta}}(x) < -1$ for some $x \in [x_{i_\ell}, x_{i_{\ell+1}}]$.*

*Proof.* Assume that $\mathcal{N}_{\boldsymbol{\theta}}(x) < -1$ for $x \in [x_{i_\ell}, x_{i_{\ell+1}}]$. Since $\mathcal{N}_{\boldsymbol{\theta}}(x_{i_\ell}) = \mathcal{N}_{\boldsymbol{\theta}}(x_{i_{\ell+1}}) = -1$, then we have $x \in (x_{i_\ell}, x_{i_{\ell+1}})$. Now, since $\mathcal{N}_{\boldsymbol{\theta}}(x_{i_\ell}) = \mathcal{N}_{\boldsymbol{\theta}}(x_{i_{\ell+1}}) = -1$ and $\mathcal{N}_{\boldsymbol{\theta}}(x) < -1$ for some $x \in (x_{i_\ell}, x_{i_{\ell+1}})$, then there must be an activation point in $(x_{i_\ell}, x_{i_{\ell+1}})$ where the derivative increases. Since by Lemma E.6 there are at most two such activation points in $[x_a, x_b]$ then the lemma follows. $\quad\square$

**Lemma E.8.** *Suppose that $x_b > x_a \geq 0$ and for all $a \leq i \leq b$ we have $y_i = -1$. There are at most 5 indices $\ell \in [m-1]$ such that $\mathcal{N}_{\boldsymbol{\theta}}(x) > -1$ for some $x \in [x_{i_\ell}, x_{i_{\ell+1}}]$.*

*Proof.* Assume that $\mathcal{N}_{\boldsymbol{\theta}}(x) > -1$ for $x \in [x_{i_\ell}, x_{i_{\ell+1}}]$. Since $\mathcal{N}_{\boldsymbol{\theta}}(x_{i_\ell}) = \mathcal{N}_{\boldsymbol{\theta}}(x_{i_{\ell+1}}) = -1$, then $x \in (x_{i_\ell}, x_{i_{\ell+1}})$. If $\ell \neq m-1$ then we have $\mathcal{N}_{\boldsymbol{\theta}}(x) > -1$ and $\mathcal{N}_{\boldsymbol{\theta}}(x_{i_{\ell+1}}) = \mathcal{N}_{\boldsymbol{\theta}}(x_{i_{\ell+2}}) = -1$. Hence, the interval $(x, x_{i_{\ell+1}})$ contains a point with negative derivative, and the interval $(x_{i_{\ell+1}}, x_{i_{\ell+2}})$ contains a point with non-negative derivative. Thus, there must be an activation point in $(x, x_{i_{\ell+2}})$ where the derivative increases. Therefore, there is an activation point with increasing derivative either in the interval $(x_{i_\ell}, x_{i_{\ell+1}}]$ or in the interval $[x_{i_{\ell+1}}, x_{i_{\ell+2}})$ (and possibly in both). Thus, an interval $[x_{i_\ell}, x_{i_{\ell+1}}]$ for $\ell \neq m-1$ might contain some $x$ with $\mathcal{N}_{\boldsymbol{\theta}}(x) > -1$ only if there is an activation point

with increasing derivative in $(x_{i_\ell}, x_{i_{\ell+1}}]$ or $[x_{i_{\ell+1}}, x_{i_{\ell+2}})$. Since by Lemma E.6 there are at most two such activation points in $[x_a, x_b]$, then there are at most 4 intervals $[x_{i_\ell}, x_{i_{\ell+1}}]$ with $\ell \neq m-1$ that contain some $x$ with $\mathcal{N}_{\boldsymbol\theta}(x) > -1$. The interval $[x_{i_{m-1}}, x_{i_m}]$ might also contain such $x$. Overall, there are at most 5 indices $\ell \in [m-1]$ such that $\mathcal{N}_{\boldsymbol\theta}(x) > -1$ for some $x \in [x_{i_\ell}, x_{i_{\ell+1}}]$. $\qquad\square$

**Lemma E.9.** *Suppose that $x_b > x_a \geq 0$ and for all $a \leq i \leq b$ we have $y_i = -1$. There are at most 30 boundaries between linear regions in $[x_{i_1}, x_{i_m}]$.*

*Proof.* The proof is similar to the proof of Lemma E.5. The only difference is that here we use Lemmas E.6 and E.7 in order to conclude that there are at most 7 indices $\ell \in [m-1]$ such that $\mathcal{N}_{\boldsymbol\theta}(x) \neq -1$ for some $x \in [x_{i_\ell}, x_{i_{\ell+1}}]$, and denote the set of these indices by $R$. $\qquad\square$