# OpenReview forum: "On the Effective Number of Linear Regions in Shallow Univariate ReLU Networks: Convergence Guarantees and Implicit Bias"
_NeurIPS.cc/2022/Conference — NeurIPS 2022 Accept_

### Official Review · Reviewer_SJ3s · 2022-06-26

**Rating:** 8
**Confidence:** 4
**Soundness:** 4 excellent
**Presentation:** 3 good
**Contribution:** 4 excellent

**Summary:**

This paper studies the convergence and generalizations of two layer ReLU networks trained by gradient flow (GF). It first gives a convergence guarantee of gradient flow. Then, using this result, the author proves that the gradient flow converges to a predictor whose number of linear regions is minimized up to a constant factor. Finally, using this characterization of implicit regularization, the author shows that two layer ReLU network trained by gradient flow generalizes.

**Questions:**

(1) The results of this paper only applies to depth two ReLU networks and it seems that the main reason for this restriction is that ReLU network with bias terms is no longer homogeneous when the depth is greater than two. Does a similar result holds for homogeneous ReLU networks (the one without bias terms) of general depth? If the results in [1] could be extended to the non-homogeneous case (i.e. neural networks with bias terms), could the results of this paper be extended to ReLU networks with general depth? Indeed, as observed in [1], the generalized results of [1] still holds experimentally even if the network has bias terms.

(2) This paper gives a nice characterization of the implicit bias of ReLU networks but the same results seem to work for all architectures (I mean the underlying graph of this network). As a result, it does not study how changes in architectures could affect generalization. Is it possible to extend the results in this paper to study the relationship between architectures and generalizations? In linear case, this question has been studied for various architectures [2]. It might be interesting to have some discussions of possible fusions of results in [2] and results in this paper.


References:

[1] K. Lyu and J. Li. Gradient descent maximizes the margin of homogeneous neural networks. arXiv
preprint arXiv:1906.05890, 2019.

[2] Zhen Dai, Mina Karzand, and Nathan Srebro. Representation costs of linear neural networks:
Analysis and design. In A. Beygelzimer, Y. Dauphin, P. Liang, and J. Wortman Vaughan, editors,
Advances in Neural Information Processing Systems, 2021.

**Limitations:**

The authors addressed the limitations of this work but not social impact. However, I think this is fine since the work is theoretical.

**Strengths And Weaknesses:**

The results in this paper are new to the best of my knowledge. The result on implicit regularization of two layer ReLU networks is very interesting. The result on generalization of two layer ReLU network is significant as it helps us understand why neural network generalizes. The presentation is clear.

One weakness of this paper is that the results are limited to two layer ReLU networks. Thus, it does not explain the success of deep neural networks, which occur more frequently in practice. However, I think the results in this paper is a good beginning to understand implicit regularization and generalization of deep neural networks. Another weakness is that this paper studies gradient flow instead of gradient descent, which is more interesting. However, I still think the results are interesting enough as it is common to first prove something on gradient flow and then discretize the results.

---

> ### Author Response · Authors · 2022-07-29
> **Review Response**
>
> Thank you for the positive feedback and support.
>
>
> 1. “The results of this paper only apply…” - This is an excellent question. We hope that it might be possible to extend our analysis to homogeneous univariate networks of depth larger than 2, but there are technical difficulties that we currently do not know how to overcome. We believe that it is a tantalizing direction for future research.
>
> 2. “This paper gives a nice characterization” - We agree that considering different architectures is an interesting research direction, although there seems to be a limited number of possibilities for univariate depth-2 networks.

---

> > ### Comment · Reviewer_SJ3s · 2022-08-09
> > **Response**
> >
> > Thanks for the response!

---

### Official Review · Reviewer_bw4F · 2022-07-06

**Rating:** 7
**Confidence:** 4
**Soundness:** 4 excellent
**Presentation:** 4 excellent
**Contribution:** 3 good

**Summary:**

The paper studies the optimization dynamics and implicit bias of GF on univariate single hidden layer ReLU networks in a binary classification setting. Under some assumptions on the initialization and data distribution, the authors show that sufficiently wide networks attain, with high probability, a training error $\le \frac{1}{2n}$, where $n$ is the number of training samples. Then, the authors show that if at some time GF attains a training error smaller than $\frac{1}{n}$ then it converges to zero loss and converges in direction to a network with at most $O(r)$ linear regions. This implicit bias characterization yields a generalization bound. Finally, the authors combine their results to derive their main result that characterizes GF implicit bias and, with high probability, guarantees convergence to zero loss and a generalization bound for sufficiently wide networks under some assumptions on the initialization and data distribution. To complement these results, the authors demonstrate that without sufficient over-parametrization GF is unable to achieve population loss below some absolute constant.

**Questions:**

1. Missing citation:
Mulayoff, R., Michaeli, T. and Soudry, D., 2021. “The implicit bias of minima stability: A view from function space”. NeurIPS 2021.
2. Can you please clarify the difference between $k$ and $k’$ in Remark 2.1?
3. I think that adding an explanation for what are “breakpoints” will make the paper clearer.

**Limitations:**

The authors mentioned and justified the limitations of their results.

**Strengths And Weaknesses:**

*Strengths:*
The paper is clear and well written, and it establishes several important results, detailed above, on GF optimization dynamics and implicit bias for univariate single hidden layer ReLU networks in a binary classification setting.

*Weaknesses:*
The main weaknesses, in my opinion, are the assumptions regarding GF (in contrast to GD) and the assumption on $\sigma_h$ in the initialization, which is not trivial. However, the authors discussed these issues in the paper and despite these limitations, I still think the paper made a significant contribution.

---

> ### Author Response · Authors · 2022-07-29
> **Review Response**
>
> Thank you for your review and positive feedback.
>
>
> 1. Mulayoff et al. - Thank you for bringing this paper to our attention.
>
> 2. "Can you please clarify the difference between $k$ and $k'$ in Remark 2.1" - $k$ is the width of the student in the mild over-parameterization regime, namely $k$ does not scale with $n$. $k'$ is the width of the student in the extreme over-parameterization regime, namely $k'$ is larger than $n$. This implies that if $n\to\infty$ then $k\ll k'$ and hence the terminology mild vs. extreme over-parameterization.
>
> 3. We will explain the term "breakpoints" as suggested.

---

> > ### Comment · Reviewer_bw4F · 2022-08-09
> > **Response**
> >
> > Thank you for your response.

---

### Official Review · Reviewer_X6Ek · 2022-07-07

**Rating:** 8
**Confidence:** 3
**Soundness:** 3 good
**Presentation:** 4 excellent
**Contribution:** 3 good

**Summary:**

The paper studies two-layer neural networks with a single input dimension for binary classification with exponential or logistic loss in a teacher-student setting. The teacher network is assumed to have a fixed number of hidden neurons (hence a fixed number of linear regions), and the paper explores the type of solution that gradient flow converges to, if the student network has a (much) larger number of hidden neurons. This reveals that, no matter how many neurons the student network has, gradient flow converges to a solution with small training loss and a restricted number of linear regions.

The proof is achieved in two steps: (i) Overparameterization implies convergence to sufficiently small loss (with high probability over a certain initialization) where all samples are classified correctly (ii) Gradient Flow then converges to a solution with a limited number of linear regions that depends (linearly) on the target network width (and hence the number of linear regions of the target network).

**Questions:**

- In what way would the authors argue that their results in the extremely restricted setting (univariate, one hidden layer) can tell us something about deeper networks on more input dimension?

- Similarly, what are the main challenges to go beyond the univariate setting or deeper networks? Is it conceivable that the methods develeoped here can be extended to a more general setting?

- In the lines 360-363: Is it possible to provide an intuitive explanation why the networks converges to a weight configuration with a constant number of kinks in intervals of unchanged label?

- Please clarify line 284 :"since the number of observations in the dataset far exceeds the degrees of freedom in the trained model“:
What are the "observations" this sentence refers to, and does the "trained model" refer to the student network? I suppose that the"observations“ refer to the samples, but since no assumption on the number of samples is made, their number cannot be used as a basis for an argument ("since the number of...."). Similarly in the following sentence („since the width of the network and thus also the model’s capacity scale with them, implying a bound that explicitly depends on them“), does"them“ still refer to these observations?


**Limitations:**

The paper studies a very restricted setting (see above), but all the necessary assumptions are clearly indicated and discussed.

**Strengths And Weaknesses:**

\+ The authors follow a novel approach that depends on the number of linear regions of the teacher network, irrespective of the sample size. The implicit bias found by the authors in their setting, is geometrically understandable in the sense that the point of convergence is a network with limited number of linear regions in relation to the target network function‘s number of linear regions.

\+  Since these main results are independent of the sample size, they can then be nicely combined with a generalization bound for finite VC dimension to obtain a generalization bound also for the solution of the gradient flow algorithm.

\+  While the setting is quite limited (single input dimension, single hidden layer), the results are important for the understanding of the implicit bias of gradient flow (i.e., what solutions are favored by training with gradient flow). The techniques are interesting and novel, and they build upon two recent papers by Lyu, Li and Ji, Telegarsky.  The paper provides complete, detailed proof, which look sound and correct to me (I did not check all the proofs in the supplement in complete detail).

\+  The authors provide an additional result, which shows that overparameterization is necessary for learnability, i.e., that small loss can be achieved with high probability over the weight initialization no matter the input distribution (with labels given by a 2-layer target network). This rounds up the (simplified) take-away that sufficient overparameterization is both necessary and sufficient for learning with gradient flow (in their setting).

\+ The paper is very well-written. In particular I appreciated that the author‘s motivate the key ideas of the proofs and clarify their assumptions and limitations.

\- The main weaknesses come from the very restricted setting and certain assumptions that are (technically) necessary for the results. Firstly, the proof technique seems to strongly depend on the univariate setting (one input dimension) and the non-existence of additional hidden layers. So it remains unclear how and whether the results could generalize to deeper networks in a more general setting.  Secondly, while the assumptions in Section 2.1 are indeed very mild, the additional assumptions in the main theorems appear stronger in the sense that the network‘s under study may not be similar to networks used in practice: The student network is required to be extremely wide (much larger than the teacher network), and the initialization technique is unconventional (hidden weight samples from a normal distribution with extremely large variance and output weights sampled from a normal distribution with extremely low variance).

*****
Taken together, I consider this a very nice, carefully composed paper, where the problem of applying only to a restricted setting is largely outweighed by the novel insight.
*****

Minor additional comments:

- Lines 360-363 explain the idea of the proof, but fail to give an intuition of it: In particular, it would be helpful to (intuitively) explain why(!) the networks converge to weight configuration with a constant number of kinks in an interval where the labels do not change sign.

- The statement in the abstract that something "may already hold“ is hard to follow if no additional context is given why the result may or may not hold.

---

> ### Author Response · Authors · 2022-07-29
> **Review Response**
>
> Thank you for the positive feedback and thorough review.
>
> - "The student network is required to be extremely wide (much larger than the teacher network)" - The student's required width depends mainly on the magnitude of $\rho$. For example, when the labels are determined by $f_r$ which is defined in Eq. (25), then the student's required width is roughly $\mathcal{O}(r\log(r))$, so there are cases where the student doesn't need to be much larger than the teacher.
>
> - "The statement in the abstract..." - We will clarify this.
>
>
> Questions:
>
> - "In what way would the authors argue..." - We do not argue that our analysis in the restricted setting necessarily implies insights on the multivariate or deeper setting, however one can hope, as mentioned to a different reviewer, that there is also a certain bias towards networks with a small number of linear regions. While this is merely wishful thinking, it does provide a concrete property and an interesting question to explore in a future work.
>
> - "Similarly, what are the main challenges..." - We believe that analyzing the implicit bias in a multivariate setting is the main bottleneck for achieving such generalizations, however we are convinced that generalizing our optimization guarantees (Theorem 3.1) to the multivariate case is certainly possible, if suitable assumptions on the underlying distribution and teacher network are made. We are still not sure if and how our techniques will generalize to deeper architectures.
>
> - "In the lines 360-363..." - In our proof we show that if there are too many linear regions in an interval where the labels are similar, then it contradicts the KKT conditions. A more precise argument seems to involve much more technical details, and we will attempt to come up with a more intuitive explanation.
>
> - "Please clarify line 284:..." - This paragraph refers to the case of mild over-parameterization as explained in Remark 2.1. We assume that the sample size (i.e. number of observations) is sufficiently large and much bigger than the number of parameters in the student (i.e. the trained network), which implies a generalization bound irrespective of our implicit bias result. We will clarify this paragraph.

---

> > ### Comment · Reviewer_X6Ek · 2022-08-09
> > **Response to the authors**
> >
> > I thank the authors for their response and clarifications to my questions. As this resolves my questions to the authors, I keep my score unchanged.

---

### Official Review · Reviewer_THUX · 2022-07-13

**Rating:** 5
**Confidence:** 3
**Soundness:** 3 good
**Presentation:** 3 good
**Contribution:** 2 fair

**Summary:**

This paper studies the dynamics and implicit bias of gradient flow (GF) on two-layer univariate ReLU Networks in a binary setting. It shows that under a teacher network (with r neurons), (over-parameterized) GF is guaranteed to achieve perfect training accuracy, which learns a network that has at most O(r) linear regions. This is characterized by the paper as the implicit bias of GF, which leads to a generalization bound. In addition, the paper proves a necessary condition for successful learning with over-parameterization in terms of a lower bound on the number of neurons.

**Questions:**

1. Since the theoretical results are derived under a simplified setting, I am wondering what are the implications of your results for more realistic settings that are typically adopted for deep learning?

**Limitations:**

Yes.

**Strengths And Weaknesses:**

Strength:

1. The paper studies an important problem in deep learning theory. The provided characterizations of the underlying dynamics of gradient flow may explain fundamentally the success of gradient-based deep learning.

2. The paper is well-structured, which presents the theoretical results in a coherent and logical way: over-parameterization -> small training error -> implicit bias towards a network with small complexity -> a generalization bound.

3. The result on the sufficiency of over-parameterization for successful learning is new.


Weakness:

1. The results of this paper build upon a couple of assumptions with respect to the underlying data distribution and the neural network class. In particular, it assumes a univariate neural network setting, where the inputs have a single feature. This seems to be a strong assumption to me, which should be discussed more in the paper. The challenges of extending the presented results to a multivariate setting, such as [1], are not clear to me.

[1] GRADIENT DESCENT MAXIMIZES THE MARGIN OF HOMOGENEOUS NEURAL NETWORKS, Kaifeng Lyu & Jian Li, ICLR 2020

2. Compared with existing literature that aims to understand the dynamics of gradient descent, the technical contributions of this paper are not clearly presented. It seems that the proof techniques of this paper are based on existing known techniques.

Minor ones:

1. I understand this is a theoretical paper, but synthetic experiments which test the theoretical claims (or even better, experiments under settings that go beyond your assumptions) will largely strengthen the paper.

2. A Typo: point our that -> point out that in line 193

---

> ### Author Response · Authors · 2022-07-29
> **Review Response**
>
> Thank you for the feedback and detailed review.
>
> - "The results of this paper build upon..." and "Since the theoretical results are derived..." - We agree that our results make essential use of the assumptions on the data and the architecture, however we believe that we must first understand the univariate setting which in turn may shed more light on the more complex and interesting multivariate setting. Indeed, the univariate setting has been studied extensively in recent years (see related work section in the paper). As a technical side note, we believe that the study of the univariate case can identify certain properties that facilitate our analysis (such as counting the number of linear regions in the student) which could potentially extend to the multivariate setting, pinpointing concrete and interesting questions to explore in future work.
>
> - "Compared with existing literature..." - We believe that our technical contribution is two-fold: (i) We develop a technique which allows one to show that the objective satisfies the PL-condition locally in a neighborhood around the initialization, for network widths that are independent of the sample size. To the best of our knowledge, our paper is the first to provide such a guarantee in a setting where the output neuron's weights are being trained (see the Teacher-student setting and mild over-parameterization paragraph in the paper). (ii) Our characterization of the implicit bias indeed builds on existing works such as Lyu and Li [2019] and Ji and Telgarsky [2020]. However, our technique is based on a careful analysis of the KKT conditions that the point of convergence must satisfy, which elucidates a novel constraint on the number of linear regions such a point can have. We hope that our proof ideas will stimulate more such analyses in the future.
>
> - "I understand this is a theoretical paper..." - We agree that additional experimentation can support our theory and we will examine adding such experiments in a revised version.

---

### Meta-Review · Area_Chair_huCd · 2022-08-23

**Recommendation:** Accept
**Confidence:** Certain

**Metareview:**

Overall: The paper focuses on an end-to-end learning guarantee for gradient flow on shallow univariate neural networks in a binary classification setting.

Reviews: The paper received four reviews. 2 Strong accepts (confident and fairly confident), Accept (confident), borderline accept (fairly confident). It seems that there are at least three reviewers that will champion the paper for publication. The reviewers found the paper is clear and has a clean presentation. The findings are interesting. The authors have provided extensive answers to reviewers' comments, answering most of them successfully.

After rebuttal: A subset of the reviewers engaged in a consensus that the paper should be accepted.

Confidence of reviews: Overall, the reviewers are confident. We will put more weight to the reviews that got engaged in the rebuttal discussion period.



**Award:**

No

---

### Decision · Program_Chairs · 2022-09-14

Accept